# REP: Resource-Efficient Prompting for Rehearsal-Free Continual Learning

**Sungho Jeon, Xinyue Ma, Kwang In Kim, Myeongjae Jeon**
POSTECH
{sunghojeon,xinyuema,kimkin,mj.jeon}@postech.ac.kr

## Abstract

Recent rehearsal-free continual learning (CL) methods guided by prompts achieve strong performance on vision tasks with non-stationary data but remain resource-intensive, hindering real-world edge deployment. We introduce resource-efficient prompting (REP), which improves the computational and memory efficiency of prompt-based rehearsal-free continual learning methods while minimizing accuracy trade-offs. Our approach employs swift prompt selection to refine input data using a carefully provisioned model and introduces adaptive token merging (AToM) and adaptive layer dropping (ALD) for efficient prompt updates. AToM and ALD selectively skip data and model layers while preserving task-specific features during the learning of new tasks. Extensive experiments on multiple image classification datasets demonstrate REP's superior resource efficiency over state-of-the-art rehearsal-free CL methods.

## 1 Introduction

Continual learning (CL) trains neural network models on multiple sequential tasks, where each task may include data distributions that diverge from previously encountered data. A crucial challenge for any CL algorithm is to effectively address *catastrophic forgetting* [23]. Severe forgetting occurs when a model rapidly loses previously learned knowledge while adapting to new tasks, significantly affecting the model's reliability and accuracy on earlier tasks.

Moreover, many of today's AI services are designed for on-device scenarios to securely learn tasks locally [19, 10, 42]. In on-device CL, improving *computational efficiency* is crucial, as it directly reduces energy usage and enhances the durability of edge devices. Meanwhile, since device memory capacity often acts as a hard constraint, CL tasks that exhaust all available memory can cause system crashes due to out-of-memory errors [9]. Given typically limited memory sizes (1–8GB), achieving high *memory efficiency* is essential to enable diverse on-device deployment scenarios in practice.

In this paper, we propose REP (Fig. 1), a framework for *resource efficiency* in prompt-based rehearsal-free CL methods on frozen, pre-trained vision transformers (ViTs) [37, 38, 35, 31, 14, 13, 30, 33]. Prompts are a small set of parameters that progressively learn incoming tasks to combat forgetting. Updates to these compact prompts incur minimum data writes without significantly harming the lifespan of device storage, which typically sustains up to 10K of writes per location. This makes prompt-based rehearsal-free methods well-suited for on-device continual learning.

REP is built upon our analysis of cost-accuracy trade-offs throughout the end-to-end learning process, with two key design insights. (1) The *prompt selection* stage, which constructs a prompt subset to augment input or intermediate data is highly amenable to numerous promising options with fast approximations. (2) In contrast, the *prompt update* stage, which involves forward-backward passes over the backbone model, presents a range of optimizations with vastly different cost-accuracy trade-offs. Based on these insights, we develop three complementary techniques for both stages that trade negligible accuracy drop for high resource efficiency: (1) *random projection-based lightweight*

39th Conference on Neural Information Processing Systems (NeurIPS 2025).

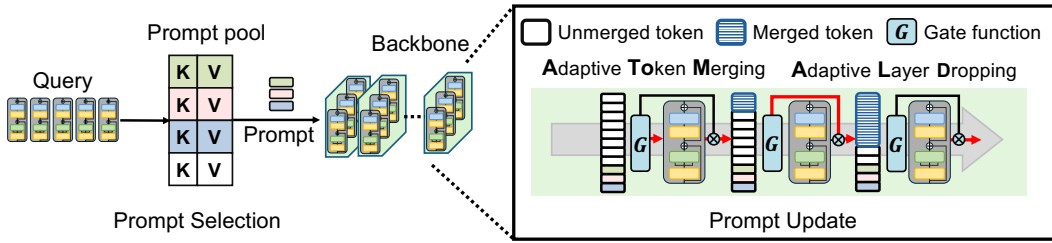

Figure 1: Overview of the proposed resource-efficient prompting (REP) algorithm for rehearsal-free CL. REP calculates query features from input samples using a lightweight surrogate model (e.g., ViT-Ti) and random projections to swiftly extract prompts from the prompt pool. These prompts are then inserted into a main backbone model (e.g., ViT-L) for training, which prioritizes model accuracy.

*surrogate model* for prompt selection, and (2) *adaptive token merging* (AToM) and (3) *adaptive layer dropping* (ALD) for prompt update. Notably, AToM and ALD *non-uniformly* skip parts of the data and model layers to reduce training costs while preserving critical task-specific knowledge. These approaches are inspired by our analysis (Fig. 2) and prior works [37, 28] demonstrating that shallower layers in pre-trained ViTs are more sensitive to new tasks than deeper layers.

Experimental results demonstrate that REP considerably reduces training time (up to 51%) and memory usage (up to 41%), with only a marginal accuracy drop, when applied to seven state-of-the-art prompt-based CL methods across three backbone models of varying sizes. As the proposed techniques are complementary in optimizing resource efficiency, we further evaluate them by integrating them into two distinct ViT-based, rehearsal-free CL methods that do not rely on prompting. In these settings, REP achieves a 37–48% reduction in training time and up to a 48% decrease in memory usage, confirming that its efficacy can extend well beyond prompt-based CL scenarios.

## 2 Related work

**Class-incremental learning.** Our work focuses on a class-incremental learning (CIL) setting, where each task introduces new classes within the same domain, but task IDs are unavailable during inference [8]. In CNN-based approaches, *rehearsal-based* methods that replay stored samples remain dominant due to their strong performance [1, 4, 5, 26, 29]. Recent methods such as BudgetCL [27] and CarM [20] adopt data-driven strategies that outperform optimization-based methods under computational constraints. MEMO [43] trades stored samples for task-specific layers when beneficial. We compare our prompt-based approach with these methods and show superior performance under resource constraints.

**Continual learning for the edge.** Recent efforts in on-device learning have emphasized memory and energy efficiency, primarily outside the CL domain [9, 34, 36]. A few studies have extended CL to edge devices: Hayes and Kanan [10] investigated online CL with CNNs for embedded systems; Kwon et al. [19] analyzed rehearsal vs. regularization methods in terms of storage, compute, and accuracy trade-offs; and Ma et al. [22] introduced Miro, a platform that dynamically configures CNN-based CL to reduce energy consumption within memory constraints. However, none of these works address the challenge of resource-efficient vision transformers (ViTs) for on-device CL, which is the primary focus of our work.

**Prompting for continual learning.** Prompting, which provides explicit instructions or queries to the model, has proven effective for ViTs in CL by enabling adaptation to new tasks while preserving prior knowledge [13, 14, 30, 31, 33, 35, 37, 38]. L2P [38] and DualPrompt [37] retrieve task-relevant prompts from a shared pool using prompt tuning and prefix tuning, respectively. CODA-Prompt [31] improves prompting capacity with attention-conditioned prompts, while OVOR [13] uses a single prompt to accelerate selection. HiDe-Prompt [33] extends supervised prompt-based CL to self-supervised settings through hierarchical optimization. ConvPrompt [30] employs convolution-based, layer-wise prompts to improve knowledge transfer with negligible overhead. LAE [7] interprets prompts as a form of parameter-efficient fine-tuning, incorporating modules such as adapters or

LoRA. As shown in Section 4, our techniques can be applied to these methods to enhance resource efficiency without significant accuracy loss.

# 3 REP: Resource-Efficient Prompting

Prompt-based rehearsal-free CL achieves strong performance by mitigating forgetting without relying on replay. However, it remains resource-intensive during the prompt selection and update stages. In particular, prompt updates require full forward and backward passes, which are costly for large backbones, limiting their applicability on resource-constrained devices.

## 3.1 Motivation: challenges in prompt selection and update

**Prompt selection** operates a neural network $f_{\text{query}}$ to compute a query feature $q(x_i^j) \in \mathbb{R}^D$ for a given input $x_i^j$ from task $j$, It then selects the prompt $p^*$ from the prompt pool $P$ that maximizes cosine similarity with the query feature:

$$p^* = \underset{p_k \in P}{\operatorname{argmax}} \frac{\langle q(x_i^j), p_k \rangle}{\|q(x_i^j)\| \|p_k\|}. \tag{1}$$

In practice, $f_{\text{query}}$ is often the same as the main backbone $f_{\text{update}}$, and each query adds an extra forward pass per sample. For large models such as ViT-L, this results in up to 28% increase in computation time (see Table 2).

**Prompt update** refines learnable parameters for effective training and adaptation by combining the classification loss $L_{\text{class}}$, a prompt-specific loss $L_{\text{prompt}}$, and an auxiliary loss $L_{\text{aux}}$ during task $j$:

$$L = L_{\text{class}}\big(f_{\text{update}}(x_i^j),\, y_i^j\big) \;+\; \epsilon_1\, L_{\text{prompt}}\big(p^*,\, q(x_i^j)\big) \;+\; \epsilon_2\, L_{\text{aux}}, \tag{2}$$

where $f_{\text{update}}$ denotes the main backbone, and $\epsilon_1$ and $\epsilon_2$ control the influence of the prompt-specific and auxiliary losses. Different methods instantiate this loss differently. For example, [37, 38] omit the auxiliary loss by setting $\epsilon_1 = 1, \epsilon_2 = 0$, whereas [31, 33] tune both coefficients.

Although most backbone layers remain frozen, each mini-batch still incurs full forward and backward passes through the network, requiring storage of intermediate activations. As a result, large backbones such as ViT-L exhibit substantial memory usage and prolonged training time (see Table 1).

## 3.2 Proposed solutions: REP

REP is built on two core insights:

1. The prompt selection stage can tolerate approximations: high retrieval quality can be achieved without relying on the full-capacity backbone.
2. Not all layers in a frozen backbone contribute equally to adaptation, allowing selective computation during prompt updates.

To realize these insights, REP introduces three complementary techniques: a lightweight surrogate model for efficient prompt selection, and adaptive token merging (AToM) and adaptive layer dropping (ALD) for efficient prompt updates.

### 3.2.1 Guiding prompt selection with a lightweight surrogate model

To reduce the computational cost of prompt selection, we use a lightweight surrogate model $f_{\text{efficient}}$ with reduced depth and width. Instead of reusing a large backbone such as ViT-L, we adopt a compact pre-trained ViT-Ti model, which effectively captures essential representations. Given an input $x_i^j$, $f_{\text{efficient}}$ produces a low-dimensional query feature $q_{\text{efficient}}(x_i^j) \in \mathbb{R}^d$ (where $d \leq D$). We then apply a fixed, non-trainable random projection $\phi$ [24] to map this feature back to the original $D$-dimensional space for prompt selection:

$$p^*_{\text{efficient}} = \underset{p_k \in P}{\operatorname{argmax}} \frac{\langle \phi(q_{\text{efficient}}(x_i^j)), p_k \rangle}{\|\phi(q_{\text{efficient}}(x_i^j))\| \|p_k\|}. \tag{3}$$

Empirically, this strategy preserves around 97% of the representational similarity to a ViT-L-based query model, as measured by centered kernel alignment [17]. Despite residing in a low-dimensional space, $p^*_{\text{efficient}}$ effectively approximates the large model's representations, substantially reducing computational cost (see Table 2).

### 3.2.2 Analysis of the frozen backbone during prompt update

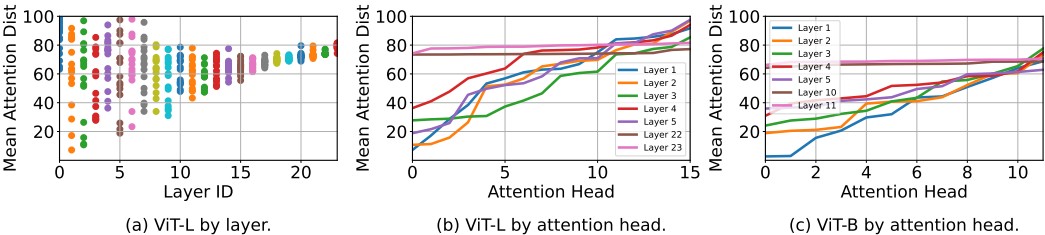

(a) ViT-L by layer.    (b) ViT-L by attention head.    (c) ViT-B by attention head.

Figure 2: Mean attention distances for frozen blocks along **(a)** layers and **(b/c)** attention heads. We run the first task of Split ImageNet-R. **(a/b)** L2P with ViT-L, and **(c)** DualPrompt with ViT-B.

A key insight of REP is that not all frozen blocks contribute equally to the final loss $L$. To validate this, we analyze the feature representations of pre-trained transformer blocks with prompts using *mean attention distance*, a metric previously used to study layer-wise representations in ViT models during pre-training [6, 28].

For a given attention head within a frozen block, let $(x_q, y_q)$ denote the position of a query patch, and $(x_i, y_i)$ the positions of the patches the head attends to, with corresponding attention weights $a_i$. The distance $d_i$ between patches—typically defined as Euclidean or pixel distance—is given by $d_i = (x_i - x_q)^2 + (y_i - y_q)^2$. We then compute the weighted mean distance as $\frac{\sum_i a_i \cdot d_i}{\sum_i a_i}$. Following [6, 28], a smaller distance indicates attention focused on a narrow region (local information), while a larger distance suggests more distributed attention across the input (global information). We measure the average mean attention distance of L2P with the ViT-L backbone along two dimensions: layer ID and attention head, during adaptation to a new task in Split ImageNet-R (10 tasks).

As shown in Fig. 2(a), shallower layers exhibit a wider range of attention distances and tend to focus on both localized and global regions. In contrast, deeper layers consistently produce higher values, reflecting a shift toward more global contextual information. This pattern aligns well with prior findings in [6, 28]. A closer look at a few selected layers in Fig. 2(b) further reveals that attention distances across heads vary more widely in shallow layers than in deeper ones, confirming that shallower layers capture more important representations of the input.

A similar trend is observed when using DualPrompt with ViT-B in Fig. 2(c). Unlike L2P, DualPrompt attaches prompts to the self attention layer of multiple transformer blocks, rather than only in the input sequence. This suggests that the feature representations of the pre-trained model, which balance local and global information, may not be greatly affected by the prompt method in use.

### 3.2.3 Adaptive token merging (AToM)

We explain how we bring the above insights into practice through two compute-skipping techniques. We first consider the data-efficient compute-skipping method via token merging. Conventional token merging (ToMe) [2] reduces the number of tokens by *uniformly* merging $n$ redundant tokens per layer, controlled by a fixed scheduler $r$. The scheduler function $r(l) \rightarrow n$ is applied to each layer $l$, and according to [2], it merges all tokens, including the prompts added by the token selection process. However, insights from Fig. 2 and our additional analysis highlight two major problems.

First, there is a *loss of task-specific information*. The prompt tokens in CL carry essential task-related information. However, ToMe indiscriminately combines these prompt tokens with non-prompt tokens, diminishing their intrinsic value. According to our empirical data in Fig. 3, this approach can cause gradient explosions in the prompt tokens even with gradient clipping, leading to learning instability.

Second, there is a *lack of layer-specific adaptability*. ToMe does not account for the disparity between shallow and deep layers, treating the importance of all layers uniformly. Therefore, there is a risk of

excessive loss of valuable information in shallow layers, which are mainly responsible for adaptability to diverse sequential tasks.

Our *adaptive token merging* (AToM; Algorithm 1) addresses the loss of task-specific information by excluding prompt tokens during token merging, thereby maintaining their specificity. To enhance the effect of prompts and mitigate catastrophic forgetting, AToM uses a new progressive scheduler $r'(l) \rightarrow n'$, which dynamically adjusts the number of tokens to merge based on layer depth, as follows:

$$r'(l) = \min(\delta \times (l - 1), r_{\max}), \qquad (4)$$

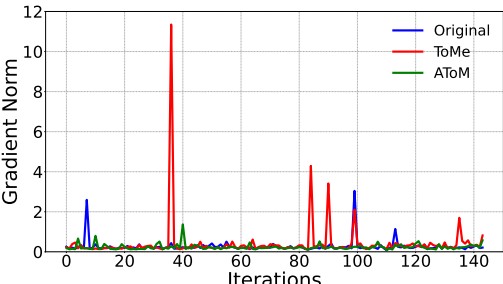

Figure 3: The norm of gradient with respect to the prompt during training Split ImageNet-R (10 tasks) when AToM and ToMe (Conventional token merging) are applied to L2P with ViT-L.

where $l$ denotes the layer index, $\delta$ is the step change in the number of tokens to merge, defined as $\frac{r_{\max}}{L-1}$ (with $L$ being the number of layers), and $r_{\max}$ is the maximum number of tokens to merge (by default, $2 \times n$). With this $r'(l)$, merging occurs more aggressively in deeper layers than in shallow ones, preserving important task-related representations.

### 3.2.4 Adaptive layer dropping (ALD)

Inspired by insights from Fig. 2 and prior work on progressive layer dropping (PLD) [41] in the NLP domain, we propose *adaptive layer dropping* (ALD; Algorithm 2) with two key features: (1) the dropping schedule considers both temporal and spatial dimensions, and (2) it manages to drop layers non-uniformly to preserve critical representation features in shallower layers.[1] On the contrary, PLD considers only the temporal aspect and does not differentiate between layers when dropping. This results in poorer performance compared to ALD, as shown in Table 3.

ALD prioritizes retaining shallow layers that contain richer information essential for model performance, especially after token merging. Thus, instead of operating on its schedule parameters, ALD leverages feedback from AToM, specifically the count of merged tokens at each layer, to guide layer-dropping decisions. The layer-keeping probability $\theta_{t,l}$ is defined as:

$$\theta_{t,l} = \left( \alpha(l) \times ((1 - \bar{\theta}) \exp(-\gamma \cdot t) + \bar{\theta}) \right), \qquad (5)$$

where $\theta_{t,l}$ is the probability of keeping layer $l$ at time step $t$, $\bar{\theta}$ is the minimum probability, $\gamma$ controls the decay rate, and $\alpha(l)$ is the adjustment factor for layer $l$, defined as:

$$\alpha(l) = \begin{cases} \alpha & \text{if } (n(l) - n'(l)) \geq \tau \\ 1 & \text{if } (n(l) - n'(l)) < \tau. \end{cases} \qquad (6)$$

$\alpha(l)$ quantifies the degree of token merging performed by AToM in layer $l$. $n(l)$ represents the original number of tokens, and $n'(l)$ denotes the number of tokens remaining after merging. When the number of merged tokens surpasses the threshold $\tau$, ALD adjusts the layer-dropping probability based on $\alpha$. Since deeper layers tend to merge more tokens with AToM, ALD is more likely to exceed $\tau$ in deeper layers and drop more aggressively. These parameters should be tuned to balance between efficiency and the preservation of nuanced information contained in the merged tokens. We set $\alpha$ to 0.9, with $\tau$ set to 16 for ViT-L, 12 for ViT-B, and 8 for ViT-Ti, respectively.

**Comparison with various layer-dropping strategies.** We compare ALD against multiple layer-dropping strategies that skip computation across different portions of backbone layers. We report the training wall-clock GPU time and final average accuracy in Fig. 4, using L2P with the ViT-L backbone on Split ImageNet-R (10 tasks). Specifically, Top-layer Drop and Bottom-layer Drop statically drop the first and last 25% of layers, respectively. Random Drop randomly skips 25% of layers across the network, while Stochastic Depth applies a linear decay schedule over Random Drop. For reference, we include No Drop, which performs no layer dropping and yields 75% accuracy. To account for the stochastic nature of compared layer dropping methods, such as Stochastic Depth, we

---

[1]The spatial dimension refers to model layers, each processing input features at different levels of abstraction.

measure end-to-end wall-clock GPU time. Top-layer Drop performs the worst, achieving only 30% accuracy. Bottom-layer Drop (63%), Random Drop (70%), and Stochastic Depth (72%) also face noticeable accuracy degradation despite reduced GPU time.

In contrast, ALD maintains 75% accuracy—on par with No Drop—while achieving GPU time savings comparable to Random Drop. This highlights that ALD's non-uniform strategy accounts for each layer's essential spatial representations more effectively. REP triggers AToM and ALD for each task insertion. Much like our prompt-selection optimization, AToM consistently enhances time and memory efficiency. In contrast, ALD solely contributes to reducing time costs since it operates with the layer-keeping probability $\theta_{t,l}$ initialized at 1.0, i.e., no layer dropping.

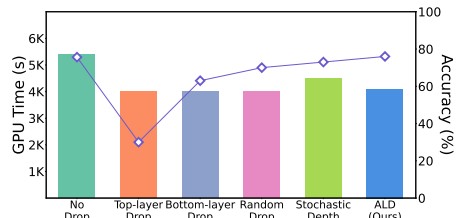

Figure 4: Comparing various layer-dropping strategies using L2P with the ViT-L backbone on Split ImageNet-R (10 tasks). The bar and marker are GPU time and final average accuracy, respectively.

### 3.2.5 Discussion

**Alternative adaptation approaches.** Prompt-based adaptation represents an effective paradigm for continual learning, particularly in resource-constrained settings, owing to its modularity and small parameter footprint. Unlike full fine-tuning, prompting keeps the backbone entirely frozen while allocating a small number of learnable parameters per task. This not only preserves prior knowledge and prevents catastrophic forgetting but also drastically reduces memory usage, which is a critical factor for edge devices. In comparison to alternatives such as LoRA, RanPAC, and adapter tuning, prompting exhibits a superior balance between efficiency and performance, as demonstrated by our integration of REP into multiple state-of-the-art prompt-based CL methods.

**Compact backbone networks (e.g., CNNs).** Although CNNs are generally more compact than vision transformers (ViTs), they consistently underperform in the CL setting, particularly under tight resource constraints. Our preliminary studies in Section D demonstrate that state-of-the-art CNN-based CL methods, such as BudgetCL [27] and MEMO [43], achieve significantly lower accuracy compared to ViT-based approaches.

## 4 Experiments

We focus on the popular disjoint-CIL setup, where each task comprises a distinct set of non-overlapping classes and samples from old classes are not given in future tasks [29, 4, 8]. In this section, we demonstrate that (1) REP enhances resource efficiency in prompt-based rehearsal-free CL across various backbone model sizes and (2) its core components, ALD and AToM, are applicable even when CL methods do not employ prompting.

**Data generation.** To organize task streams, we use three image classification datasets: CIFAR-100 (100 classes) [18], ImageNet-R (200 classes) [12], and PlantDisease (38 classes) [25]. Out of these datasets, ImageNet-R is known for exhibiting much higher intra-class variability and an uneven class-size distribution among images. We divide CIFAR-100 and ImageNet-R into 10 tasks to create **Split CIFAR-100** (i.e., 10 classes per task) and **Split ImageNet-R** (i.e., 20 classes per task), respectively [38, 37, 31]. For PlantDisease, we drop 3 plant disease classes with very few images and organize the remaining 35 classes into 7 tasks to create **Split PlantDisease** (i.e., 5 classes per task). We also present additional results with varying task lengths (e.g., 5 tasks and 20 tasks), and an additional Split CUB-200 dataset [32] in Section F and Section G, respectively.

**Methods.** We employ **L2P** [38], **DualPrompt** [37], **CODA-Prompt** [31], **HiDe-Prompt** [33], **ConvPrompt** [30], **LAE** [7], and **OVOR** [13] as representative prompting methods. These methods capitalize on ImageNet pre-trained models as backbones: ViT-L (307M), ViT-B (86M), and ViT-Ti (5.8M)—ViT-Ti is *one of the smallest vision transformer models* to our knowledge. Detailed settings for all methods are provided in Section B.

**Hardware and metrics.** We use an NVIDIA RTX 3090 to cover a wide range of memory capacities while maintaining consistent computational power. Specifically, we limit the GPU memory to emulate

Table 1: Accuracy and computational cost of all competing methods on three datasets. We report the final average accuracy, iteration time, and memory usage, both without and with REP. Iteration time and memory usage are presented as absolute values. For the results without REP, parentheses indicate how many times higher the value is relative to its counterpart with REP.

| Model | Method | Final average accuracy (↑) | | | | | | Iter. time (ms) | | Mem. (GB) | |
|---|---|---|---|---|---|---|---|---|---|---|---|
| | | Split CIFAR-100 | | Split ImageNet-R | | Split PlantDisease | | | | | |
| | | w/o REP | w/ REP | w/o REP | w/ REP | w/o REP | w/ REP | w/o REP | w/ REP | w/o REP | w/ REP |
| ViT-L | L2P | 88.2±0.3 | 87.0±1.3 | 75.6±1.0 | 75.3±1.5 | 75.9±3.1 | 81.1±3.9 | 447(×1.9) | 240 | 6.5(×1.4) | 4.5 |
| | DualPrompt | 86.3±0.3 | 85.1±1.3 | 71.2±0.6 | 70.6±0.9 | 75.1±1.1 | 78.2±2.9 | 424(×2.0) | 208 | 5.9(×1.4) | 4.3 |
| | CODA-Prompt | 85.4±0.4 | 84.3±1.7 | 74.4±0.6 | 73.5±1.9 | 75.0±2.0 | 74.1±2.7 | 568(×1.3) | 441 | 13.2(×1.2) | 11.0 |
| | HiDe-Prompt | 93.4±0.3 | 92.8±1.1 | 78.7±0.2 | 78.0±1.2 | 98.6±0.4 | 98.3±0.8 | 413(×1.8) | 227 | 7.3(×1.2) | 6.3 |
| | ConvPrompt | 89.2±0.5 | 89.0±0.9 | 79.1±0.1 | 78.5±0.8 | 98.4±0.6 | 98.6±0.7 | 560(×1.3) | 417 | 5.4(×1.3) | 4.1 |
| | LAE-Prefix10 | 84.9±1.3 | 84.5±1.5 | 70.5±1.9 | 70.2±1.7 | 73.3±0.5 | 72.9±0.9 | 185(×1.1) | 170 | 5.1(×1.1) | 4.8 |
| | OVOR | 86.2±0.6 | 85.8±0.9 | 75.4±0.7 | 74.3±1.3 | 75.4±0.8 | 74.9±1.0 | 467(×1.3) | 362 | 7.5(×1.1) | 6.8 |
| | UpperBound | 94.1±0.1 | – | 85.2±0.2 | – | 99.6±0.3 | – | 427 | – | 9.8 | – |
| ViT-B | L2P | 84.5±0.8 | 83.4±1.8 | 59.4±0.8 | 58.7±1.2 | 64.4±2.6 | 64.0±2.4 | 143(×1.4) | 102 | 2.3(×1.2) | 2.0 |
| | DualPrompt | 85.1±0.2 | 84.9±0.9 | 68.1±0.1 | 67.0±1.0 | 78.0±1.5 | 77.2±1.9 | 133(×1.3) | 101 | 2.1(×1.2) | 1.8 |
| | CODA-Prompt | 83.3±0.9 | 82.1±1.9 | 59.0±0.3 | 58.3±1.8 | 71.4±0.8 | 70.9±2.7 | 164(×1.1) | 151 | 6.9(×1.2) | 5.7 |
| | HiDe-Prompt | 88.5±0.7 | 89.2±1.6 | 64.5±0.5 | 64.4±1.0 | 94.9±0.4 | 94.5±0.8 | 130(×1.7) | 76 | 4.1(×1.7) | 2.4 |
| | ConvPrompt | 87.6±0.5 | 87.0±1.4 | 67.6±0.6 | 67.1±1.2 | 94.3±0.8 | 94.5±0.9 | 381(×1.4) | 279 | 2.2(×1.1) | 2.0 |
| | LAE-Prefix10 | 83.5±0.8 | 84.2±1.3 | 61.3±0.6 | 61.0±0.9 | 70.8±0.6 | 70.4±1.1 | 57(×1.1) | 50 | 1.9(×1.1) | 1.8 |
| | OVOR | 84.2±0.7 | 83.9±1.0 | 61.0±0.6 | 60.6±1.0 | 67.2±0.5 | 66.9±0.9 | 134(×1.1) | 123 | 3.1(×1.1) | 3.0 |
| | UpperBound | 92.0±0.2 | – | 81.4±0.1 | – | 99.6±0.4 | – | 143 | – | 3.2 | – |
| ViT-Ti | L2P | 60.3±0.2 | 59.3±1.7 | 41.2±0.7 | 40.3±1.0 | 56.1±2.9 | 55.2±2.9 | 34(×1.1) | 30 | 0.5(×1.1) | 0.5 |
| | DualPrompt | 62.9±0.5 | 62.5±0.9 | 43.6±0.8 | 42.7±1.0 | 64.0±1.0 | 62.9±1.9 | 34(×1.1) | 31 | 0.4(×1.1) | 0.4 |
| | CODA-Prompt | 67.0±0.4 | 65.9±1.8 | 49.3±0.8 | 48.2±2.0 | 66.8±0.8 | 65.5±2.8 | 36(×1.2) | 30 | 2.8(×1.1) | 2.8 |
| | HiDe-Prompt | 72.2±0.5 | 72.5±1.0 | 44.8±0.8 | 45.2±1.4 | 92.6±0.7 | 93.3±1.5 | 23(×1.2) | 20 | 0.5(×1.1) | 0.4 |
| | ConvPrompt | 69.1±0.8 | 68.5±1.7 | 49.8±0.8 | 49.1±1.9 | 92.4±0.6 | 93.4±0.9 | 85(×1.9) | 45 | 0.5(×1.1) | 0.5 |
| | LAE-Prefix10 | 61.3±0.6 | 61.3±1.4 | 49.7±0.8 | 49.6±1.3 | 66.8±0.7 | 66.0±1.5 | 15(×1.3) | 12 | 0.4(×1.1) | 0.4 |
| | OVOR | 67.7±0.8 | 67.0±1.3 | 49.8±0.7 | 49.3±1.2 | 70.3±0.6 | 71.6±1.7 | 35(×1.1) | 32 | 1.0(×1.1) | 0.9 |
| | UpperBound | 83.0±0.1 | – | 77.4±0.2 | – | 98.0±0.9 | – | 42 | – | 0.6 | – |

resource-constrained devices, ranging from 1GB (compatible with Raspberry Pi) to 16GB (compatible with NVIDIA Jetson AGX Xavier). For computational efficiency, we measure *iteration time*, which is linearly correlated with GPU energy usage [22]. We adopt the *final average accuracy* (higher is better) and *forgetting* (lower is better) from various previous works [37, 38, 31, 33]. In Section M, we further evaluate REP directly on NVIDIA Jetson TX2 with 8GB of memory, although it is applicable beyond this specific device.

## 4.1 Main results

Table 1 presents the efficacy of REP when integrated into prompt-based CL methods across various ViT backbones (ViT-L, ViT-B, and ViT-Ti) and datasets. For a direct comparison of these methods under their original setups, please refer to Section C.

**Resource efficiency.** Integrating REP into existing prompt-based CL methods substantially reduces computational overhead and memory usage across all model configurations. As shown in Table 1, baseline methods without REP consume up to $1.4\times$, $1.7\times$, and $1.1\times$ more GPU memory than those with REP for ViT-L, ViT-B, and ViT-Ti, respectively. Although the ViT backbones remain frozen, these models still perform backward passes to optimize prompts and classification heads, requiring memory to store large intermediate activations. REP mitigates these memory demands by selectively merging tokens (AToM) and adaptively dropping layers (ALD), thus reducing the memory footprint of intermediate data.

In addition, baseline methods without REP use up to $2.0\times$, $1.7\times$, and $1.9\times$ more training time than those with REP for ViT-L, ViT-B, and ViT-Ti, respectively. These improvements in computational efficiency directly translate into proportionally lower energy consumption. The gains are more marked in ViT-L, as its larger size offers greater room for optimizing computation cost.

**Accuracy.** While all baseline methods were originally designed and evaluated on a ViT-B backbone, our study broadens their applicability by exploring cost-accuracy trade-offs in on-device CL across a diverse range of ViT architectures. When augmented with our techniques, these methods achieve notable gains in resource efficiency with only marginal accuracy degradation: 0.0–1.2% on Split CIFAR-100, 0.1–1.1% on Split ImageNet-R, and 0.0–0.8% on Split PlantDisease. In some cases,

accuracy even improves with REP; for example, the accuracy of L2P with ViT-L on Split PlantDisease increases from 75.9% to 81.1% when combined with REP. For completeness, the forgetting metric is reported in Section I.

## 4.2 Ablation studies

We validate our proposed techniques and the chosen hyperparameters, primarily using REP integrated into L2P with the ViT-L backbone (denoted as REP-L2P) on Split ImageNet-R (10 tasks). See Section E for additional results.

**Component ablation.** In Table 2, we ablate the components of REP to assess their individual contributions. Each row reports performance when the corresponding component is removed. All components contribute substantially to reducing computation time and memory usage, with ALD affecting only computation time, as expected.

Table 2: Ablation study of REP's components, demonstrating their contributions to both resource efficiency and accuracy.

| Ablated components | Acc. ($\uparrow$) | Fgt. ($\downarrow$) | Iter. time (ms) | Mem. (GB) |
|---|---|---|---|---|
| REP-L2P | 75.3±1.5 | 3.6±0.3 | 240 | 4.5 |
| w/o (AToM + ALD) | 74.8±0.8 | 4.0±0.8 | 349 | 5.5 |
| w/o ($f_{efficient}$ + ALD) | 74.9±0.8 | 3.5±0.8 | 419 | 5.6 |
| w/o ($f_{efficient}$ + AToM) | 74.5±0.8 | 3.6±0.7 | 401 | 6.5 |
| w/o ALD | 74.2±0.8 | 4.0±0.8 | 270 | 4.5 |
| w/o AToM | 74.5±0.4 | 3.4±0.5 | 303 | 5.5 |
| w/o $f_{efficient}$ | 74.6±0.4 | 2.6±0.5 | 326 | 4.8 |

**Algorithm validation.** Conventional token merging (ToMe) [2] and layer dropping (PLD) [41] are specifically designed to accelerate the training of transformer-based models across various domains. To validate the importance of incorporating our adaptive techniques instead in CL, we first evaluate how REP performs in case AToM and ALD are replaced with ToMe and PLD, respectively. The results are presented in (1) of Table 3. Although applying ToMe or PLD improves resource efficiency over the native L2P baseline in Table 1, it results in an accuracy drop of 5.1% or 2.0%, respectively, compared to using our techniques.

**Surrogate prompt selection validation.** We further validate the effectiveness of our lightweight surrogate-based prompt selection method by comparing it with an alternative that uses the CLS token from intermediate layers of ViT-L (layers 4, 8, and 12). As shown in (2) of Table 3, while using intermediate layers eliminates the need for a separate surrogate, it substantially increases training time, since even a partial ViT-L forward pass is more computationally demanding than running the full surrogate model. Accuracy also drops, with

Table 3: Effect of using (1) conventional acceleration methods, (2) intermediate layer-based prompt selection method, and (3) random drop with diverse drop ratios, compared to our proposed methods.

| Method | Acc. ($\uparrow$) | Fgt. ($\downarrow$) | Iter. time (ms) | Mem. (GB) |
|---|---|---|---|---|
| *(1) Algorithm validation* | | | | |
| REP-L2P | 75.3±1.5 | 3.6±0.3 | 240 | 4.5 |
| w/ ToMe | 70.2±0.7 | 2.6±0.9 | 275 | 3.7 |
| w/ PLD | 73.3±0.7 | 3.9±0.7 | 259 | 4.5 |
| *(2) Surrogate prompt selection validation* | | | | |
| Surrogate | 75.3±1.5 | 3.6±0.3 | 240 | 4.5 |
| Interm. (4) | 74.0±0.8 | 3.9±0.9 | 264 | 4.3 |
| Interm. (8) | 74.3±0.7 | 3.5±0.9 | 285 | 4.3 |
| Interm. (12) | 74.6±0.7 | 2.6±0.8 | 307 | 4.3 |
| *(3) Comparison with FIM-based token merging* | | | | |
| AToM | 74.5±0.8 | 3.5±0.8 | 419 | 5.6 |
| FIM-based | 74.7±0.9 | 3.8±0.4 | 264 | 6.2 |
| *(4) Comparison with random drop* | | | | |
| ALD | 75.8±0.5 | 3.5±0.7 | 401 | 6.5 |
| Random-20% | 71.7±0.4 | 5.8±0.8 | 409 | 6.5 |
| Random-25% | 70.6±0.4 | 6.4±1.1 | 398 | 6.5 |
| Random-30% | 68.8±0.6 | 7.1±1.5 | 385 | 6.5 |
| Random-50% | 65.4±0.8 | 9.4±1.8 | 363 | 6.5 |

shallower-layer cases showing greater degradation, likely due to their limited access to global context.

**Comparison with FIM-based token merging.** We also investigate whether additional gradient information, e.g. through the Fisher Information Matrix (FIM) [16], could serve as an alternative to AToM. Specifically, we compare AToM with a FIM-based, gradient-informed merging strategy using L2P with ViT-L on Split ImageNet-R (10 tasks), as shown in (3) of Table 3. Although this approach yields modest accuracy gains (0.1–0.2%), it incurs roughly a 10% increase in training time and memory usage. By contrast, AToM preserves prompt gradients without such overhead, offering a more balanced trade-off between efficiency and forgetting mitigation. This result does not point to a fundamental limitation of FIM-based merging, but rather underscores the difficulty of achieving practical efficiency improvements in its current form.

**Comparison with random drop across varying aggressiveness levels.** To evaluate ALD against Random Drop with varying aggressiveness, we applied dropping rates of 20%, 25%, 30%, and 50% using L2P with ViT-L on on Split ImageNet-R (10 tasks). As shown in (4) of Table 3, higher drop rates reduce training time below that of ALD but lead to substantial accuracy degradation (65–71%),

making them impractical for continual learning. In contrast, ALD dynamically adjusts the drop ratio, maintaining resource efficiency while minimizing accuracy loss.

**AToM and ALD intensity.** Table 4 shows the effects of varying intensities of AToM and ALD, focusing on the number of merged tokens ($n$) in AToM and the keep ratio ($\theta$) in ALD. AToM appears to maintain stable accuracy trends even as more tokens are merged. In contrast, in the case of ALD, a lower keep ratio appears to improve resource efficiency by reducing training time. However, it can markedly impair model accuracy if the ratio is too low. Overall, when used with carefully selected hyperparameters, AToM and ALD can effectively balance resource efficiency and accuracy. Notably, forgetting remains stable when using the default settings ($n = 8$ and $\theta = 0.5$). In Section E, we extend the above study to two additional prompt-based CL methods, including HiDe-Prompt [33] and ConvPrompt [30].

Table 4: REP over varying # of merged tokens ($n$) and % of keep ratio ($\theta$).

| $n$ (w/ $\theta$=0.5) | Acc. (↑) | Fgt. (↓) | Iter. time (ms) | Mem. (GB) |
|---|---|---|---|---|
| 1 | 75.5±0.7 | 3.5±0.7 | 268 | 5.5 |
| 2 | 75.2±0.7 | 3.7±0.7 | 265 | 5.5 |
| 4 | 75.3±0.3 | 3.3±0.5 | 256 | 5.2 |
| 6 | 75.1±0.6 | 3.8±0.8 | 253 | 4.8 |
| 8 | 75.3±1.5 | 3.6±0.3 | 240 | 4.5 |
| 10 | 73.6±1.7 | 4.7±1.1 | 228 | 4.1 |

| $\theta$ (w/ $n$=8) | Acc. (↑) | Fgt. (↓) | Iter. time (ms) | Mem. (GB) |
|---|---|---|---|---|
| 0.1 | 72.9±1.4 | 4.4±1.5 | 217 | 4.5 |
| 0.2 | 73.2±1.2 | 4.4±1.2 | 223 | 4.5 |
| 0.3 | 74.4±1.2 | 4.1±0.8 | 228 | 4.5 |
| 0.4 | 74.7±1.0 | 3.9±0.5 | 235 | 4.5 |
| 0.5 | 75.3±1.5 | 3.6±0.3 | 240 | 4.5 |
| 0.6 | 74.8±0.9 | 3.6±0.9 | 255 | 4.5 |
| 0.7 | 74.4±0.6 | 3.5±0.5 | 270 | 4.5 |
| 0.8 | 74.2±0.7 | 3.8±0.9 | 277 | 4.5 |
| 0.9 | 74.3±0.5 | 3.6±0.8 | 282 | 4.5 |

## 4.3 Additional studies

We further assess the efficacy of REP by applying it to CL methods not covered by Table 1, including two additional non-prompting rehearsal-free methods: such as SLCA [40] and RanPAC [24]. Additionally, we compare REP-L2P with two adapter-based methods, including Online-LoRA [39] and ADAM [15]. More discussions are provided in Section H and Section J, respectively.

### 4.3.1 Applying REP to non-prompting rehearsal-free methods

SLCA fine-tunes the entire network using distinct learning rates for the representation and fully-connected layers, whereas RanPAC employs a non-trainable random projection to the activations from the pre-trained frozen backbone. For SLCA, we retain the original hyperparameters (e.g., the number of epochs per task), and integrate ConvPrompt to enable parameter-efficient fine-tuning (PEFT), as SLCA lacks native PEFT support. We measure iteration time and memory usage, with ViT-L on Split ImageNet-R (10 tasks).

As shown in Table 5, SLCA incurs high memory overhead even with ConvPrompt, limiting its compatibility with 8GB edge devices. Applying REP reduces both training time and memory usage by up to 48%, making it operational on such resource-constrained devices. Similarly, applying REP to RanPAC reduces iteration time by up to 37% (as shown in Table 6) but does not lower memory usage, as random projections from upsampling dominate the high memory cost.

Table 5: REP on SLCA with ConvPrompt using ViT-L backbone on Split ImageNet-R.

| Method | Acc. (↑) | Iter. time (ms) | Mem. (GB) |
|---|---|---|---|
| SLCA | 77.2±1.1 | 1,311 | 9.3 |
| REP-SLCA | 78.0±1.3 | 686 | 4.8 |

Table 6: REP on RanPAC using ViT-L backbone on Split ImageNet-R.

| Method | Acc. (↑) | Iter. time (ms) | Mem. (GB) |
|---|---|---|---|
| RanPAC | 82.4±0.6 | 332 | 8.2 |
| REP-RanPAC | 82.0±1.7 | 210 | 8.2 |

### 4.3.2 Comparison with adapter-based methods

**LoRA-based method.** We analyze Online-LoRA using the ViT-L backbone on Split ImageNet-R (10 tasks). As shown in Table 7, this method introduces substantial memory overhead (>20GB), making it incompatible with the memory specifications of most edge devices. This overhead arises

Table 7: Comparison of REP with Online-LoRA using ViT-L backbone on Split ImageNet-R.

| Method | Acc. (↑) | Fgt. (↓) | Iter. time (ms) | Mem. (GB) |
|---|---|---|---|---|
| REP-L2P | 75.3±1.5 | 2.8±0.2 | 240 | 4.5 |
| Online-LoRA | 52.6±1.8 | 21.2±2.6 | 1,739 | 20.9 |

from using replay buffers and multiple forward/backward passes, which significantly inflate both training time and memory usage.

**Other adapter-based methods.** Next, we look into ADAM, a representative adapter-based CL method. Table 8 presents results using the ViT-L backbone on Split ImageNet-R (20 tasks). While ADAM variants can reduce training costs through lightweight adapters, they often suffer from large memory footprints that exceed the device limits (e.g., 8GB) or from significantly lower accuracy compared to REP-L2P.

Table 8: Comparison of REP with ADAM variants using ViT-L backbone on Split ImageNet-R.

| Method | Acc. ($\uparrow$) | Iter. time (ms) | Mem. (GB) |
|---|---|---|---|
| REP-L2P | 72.6±0.6 | 240 | 4.50 |
| VPT-Deep | 66.7±0.4 | 136 | 12.6 |
| VPT-Shallow | 64.5±0.5 | 125 | 6.9 |
| SSF | 72.2±0.2 | 147 | 22.1 |
| Adapter | 62.1±0.1 | 62 | 12.0 |

## 5   Conclusions

In this work, we introduce Resource-Efficient Prompting (REP), a framework designed to enhance the computational and memory efficiency of prompt-based rehearsal-free continual learning methods. By incorporating surrogate prompt selection, adaptive token merging and adaptive layer dropping, REP selectively streamlines the learning process while preserving task-specific features, significantly reducing resource consumption without compromising accuracy. Experiments on multiple image classification datasets demonstrate that REP achieves substantial efficiency gains over state-of-the-art ViT-based rehearsal-free methods, making it a practical solution for on-device continual learning.

We focus on a standard continual learning setting where task arrivals are predefined as part of the problem formulation. In this setting, REP is designed to operate under the given task order. Curriculum learning, which explicitly determines or optimizes the task sequence, addresses a related but distinct problem. Since our study does not include curriculum-based task scheduling, REP has not been evaluated in scenarios where the task order can be deliberately arranged. Exploring how REP could be combined with curriculum learning strategies is an interesting direction for future research. Moreover, although using a replay buffer may be considered infeasible in real-world scenarios with strict data privacy constraints, on-device CL is already viewed as a privacy-preserving approach, making rehearsal-based REP a promising direction for future investigation.

One limitation of our work is hyperparameter tuning. Following prior works, we tuned the hyperparameters through grid search with 5-fold cross-validation for each task. While this provides an automatic tuning procedure, it may limit the "plug-and-play" applicability of REP. Empirically, we observed that performance is relatively insensitive to variations in $\alpha$, making it reasonable to fix $\alpha$ = 0.9. For $\tau$, the impact is more dataset- and model-dependent, but we found a consistent positive correlation between backbone size and the effective $\tau$ value. As future work, we plan to conduct a more detailed study on hyperparameter tuning.

## Acknowledgements

This work was supported by the Institute of Information & Communications Technology Planning & Evaluation (IITP) grants (No. RS-2019-II191906, Artificial Intelligence Graduate School Program (POSTECH); No. RS-2024-00459797, Development of ML Compiler Framework for On-device AI; No. RS-2025-02304554, Efficient and Scalable Framework for AI Heterogeneous Cluster Systems; No. RS-2022-II220290, Visual Intelligence for Space-Time Understanding and Generation), the National Research Foundation of Korea (NRF) grants (No. RS-2024-00337559; No. RS-2024-00354947), and the Electronics and Telecommunications Research Institute (ETRI) grant (No. 25ZS1100, Research on High-Performance Computing to Overcome Limitations of AI), all funded by the Korean government (MSIT).

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

# A    Algorithm details

---

**Algorithm 1** Adaptive Token Merging (AToM)

---

**Input**: Initial set of all tokens $T$; Set of prompt tokens $P$;
Number of model layers $L$;
Maximum number of tokens to merge $r_{\max}$
**Initialize**: $T'_{\text{final}} \leftarrow T$;
1: **for** $l \in \{1, 2, \dots, \text{L}\}$ **do**
2:    $T_{\text{attn}} \leftarrow \text{MSA}(T'_{\text{final}})$
3:    $T_{\text{eligible}} \leftarrow T_{\text{attn}} \setminus P$
4:    $\delta \leftarrow \frac{r_{\max}}{\text{L}-1}$
5:    $n' \leftarrow \min(\delta \times (l-1), r_{\max})$
6:    $T'_{\text{merged}} \leftarrow \text{Merge}(T_{\text{eligible}}, n')$
7:    $T_{\text{concat}} \leftarrow \text{Concat}(T'_{\text{merged}}, P)$
8:    $T'_{\text{final}} \leftarrow \text{MLP}(T_{\text{concat}}, l)$
9: **end for**
10: **return** $T'_{\text{final}}$

---

---

**Algorithm 2** Adaptive Layer Dropping (ALD)

---

**Input**: Input tensor $X$; Keep ratio of layer $\theta_{t,l}$;
Number of layers $L$; Minimum ratio $\bar{\theta}$;
Decay rate $\gamma$; Spatial threshold $\tau$;
Adjustment factor $\alpha$
1: **for** $l \in \{1, 2, \dots, L\}$ **do**
2:    **if** $(n(l) - n'(l)) > \tau$ **then**
3:       $\alpha(l) \leftarrow \alpha$
4:    **else**
5:       $\alpha(l) \leftarrow 1$
6:    **end if**
7:    $\theta_{t,l} \leftarrow \alpha(l) \times \big((1 - \bar{\theta})\exp(-\gamma \cdot t) + \bar{\theta}\big)$
8:    **if** $\text{Bernoulli}(\theta_{t,l}) = 1$ **then**
9:       $X_{\text{out}} \leftarrow \text{Exec}(l, X_{\text{out}})$
10:   **else**
11:       $X_{\text{out}} \leftarrow X_{\text{out}}$
12:   **end if**
13: **end for**
14: **return** $X_{\text{out}}$

---

# B    Implementation details

Unless otherwise stated, we use the term **REP-L2P** to denote REP applied to L2P with a ViT-L backbone. Similarly, the notations **L**, **B**, and **Ti** are used to generally denote **ViT-L**, **ViT-B**, and **ViT-Ti** backbones, respectively, in the prompt update the stage of any prompt-based CL method.

## B.1    Prompt-based methods for REP

We present details on the multiple prompt-based rehearsal-free CL methods to which REP is integrated for enhancing resource efficiency.

**L2P** positions prompts at the first layer of the transformer architecture [38]. These prompts are learnable parameters that dynamically evolve with the training process. The mechanism begins with a prompt pool $P = \{p_1, p_2, ..., p_m\} \subset \mathbb{R}^{L_p \times D}$.

For a given input $x_i^j$ in task $T_i$, L2P computes a query feature $q(x_i^j) \in \mathbb{R}^D$ to select the corresponding prompt. The prompt $p^*$ is selected based on maximizing the cosine similarity with respect to the

query i.e. input data:

$$p^* = \underset{p_k \in P}{\operatorname{argmax}} \sum_{c=1}^{L_p} \frac{\langle q(x_i^j), [p_k]_{(c,:)}^\top \rangle}{\|q(x_i^j)\| \|[p_k]_{(c,:)}^\top\|}, \tag{7}$$

where $[p_k]_{(c,:)}$ is the $c$-th row of $p_k$.

The selected prompt $p^*$ is concatenated with the input embedding $z(x_i^j)$ to form the prompt-augmented input $z'(x_i^j) = [p^*; z(x_i^j)]$. The training objective of L2P balances classification loss $L_{\text{class}}$ and prompt-adjustment loss $L_{\text{prompt}}$:

$$L = L_{\text{class}}(z'(x_i^j), y_i^j) + L_{\text{prompt}}(p^*, q(x_i^j)). \tag{8}$$

**DualPrompt** leverages prompts at multiple layers of the transformer architecture [37]. It introduces a general prompt $g \in \mathbb{R}^{L_g \times D}$ and a set of task-specific prompts $E = \{e_1, e_2, ..., e_T\} \subset \mathbb{R}^{L_e \times D}$. These prompts are incorporated at specified layers in the transformer model.

For a given input $x_i^j$ in task $T_i$, the model's transformer layers $f$ are modified by attaching $g$ and $e_i$ to the layers, resulting in a prompted architecture $f_{g,e_i}$. The feature transformation $h_i^j$ for the input sample $x_i^j$ is then obtained as:

$$h_i^j = f_{g,e_i}(x_i^j). \tag{9}$$

Similar to L2P, DualPrompt optimizes $L_{\text{class}}$ and $L_{\text{prompt}}$:

$$L = L_{\text{class}}(h_i^j, y_i^j) + L_{\text{prompt}}(g, e_i). \tag{10}$$

**CODA-Prompt** decomposes learnable prompts into components and uses an attention mechanism from a pre-trained ViT model to select relevant prompts [31]. Instead of a single prompt, CODA-Prompt learns a set of prompt components $P = \{P_1, P_2, ..., P_M\}$. The final prompt $p$ is calculated as a weighted sum:

$$p = \sum_m \alpha_m P_m, \tag{11}$$

where weights $\alpha$ are determined based on the query $q(x)$ and keys $K \in \mathbb{R}^{D \times M}$:

$$\alpha = \gamma(q(x), K). \tag{12}$$

When the task changes, the current components are frozen, and a new one is added, promoting orthogonality, denoted as:

$$L_{\text{ortho}}(B) = \|BB^\top - I\|^2, \tag{13}$$

where $B$ represents $P$, $K$, or $A$ ($A$ represents the *attention vector*). The full optimization target is:

$$\min_{P^n, K^n, A^n, \phi^n} L\left(f_\phi(f_{\theta, P, K, A}(x)), y\right) + \lambda\left(L_{\text{ortho}}(P) + L_{\text{ortho}}(K) + L_{\text{ortho}}(A)\right) \tag{14}$$

where $P^n$, $K^n$, and $A^n$ are new components, and $\lambda$ balances the orthogonality loss [31].

**HiDe-Prompt** decomposes the CL objective into three parts—within-task prediction (WTP), task-identity inference (TII), and task-adaptive prediction (TAP) [33]. It manages uninstructed/instructed representations by modeling each class $c$ with approximate distributions (e.g., Gaussian means $\boldsymbol{\mu}_c$). The overall combined loss is composed with WTP loss ($\mathcal{L}_{\text{WTP}}$), TII loss ($\mathcal{L}_{\text{TII}}$), TAP loss ($\mathcal{L}_{\text{TAP}}$) and a contrastive regularization loss ($\mathcal{L}_{\text{CR}}$):

$$\mathcal{L}_{\text{HiDe}} = \mathcal{L}_{\text{WTP}}(\boldsymbol{p}_t, f_\theta) + \mathcal{L}_{\text{TII}}(\omega; \widehat{\mathcal{G}}_c) + \mathcal{L}_{\text{TAP}}(\psi; \mathcal{G}_c) + \lambda \mathcal{L}_{\text{CR}}(\boldsymbol{p}_t, \{\boldsymbol{\mu}_c\}), \tag{15}$$

where $\boldsymbol{p}_t$ is the current prompt, $\omega, \psi$ are auxiliary heads, and $\widehat{\mathcal{G}}_c, \mathcal{G}_c$ denote distributions of uninstructed/instructed representations for old classes. This hierarchical approach excels under self-supervised pre-training.

**ConvPrompt** generates layer-wise prompts via convolution on a shared embedding matrix SE [30]. For a layer $\ell$ and head $h$, it produces $M$ prompt components $\mathrm{PC}_{\ell,h,m}^K$, $\mathrm{PC}_{\ell,h,m}^V$ by convolving $\mathrm{SE}_{\ell,h}^K$ and $\mathrm{SE}_{\ell,h}^V$ with $m$th prompt generator $G_{\ell,h,m}$. The final prompt is computed by weighting these components via a projection network and prompt keys:

$$P_{\ell+1,h}^{(K)} = \sum_{m=1}^{M} s_{\ell,m}\,\mathrm{PC}_{\ell,h,m}^K, \qquad P_{\ell+1,h}^{(V)} = \sum_{m=1}^{M} s_{\ell,m}\,\mathrm{PC}_{\ell,h,m}^V. \qquad (16)$$

Here, $s_{\ell,m}$ are similarity scores derived from [CLS] embeddings, enabling a *dynamic* prompt composition with small overhead.

**OVOR** employs a single prompt throughout all tasks, in contrast to methods that maintain a large prompt pool [13]. To further refine decision boundaries, it introduces virtual outlier regularization (VOR), which generates synthetic outliers $\mathbf{v} \in \mathcal{D}_{\mathrm{Outlier}}$ to penalize overconfident predictions outside the training distribution. OVOR's loss is described as:

$$\mathcal{L}_{\mathrm{VOR}} = \mathbb{E}_{\mathbf{x} \sim \mathcal{D}^t}\Big[\mathcal{H}\big(\widehat{\mathbf{y}}^{(\mathbf{x})}, \mathbf{y}\big)\Big] \; + \; \lambda\, \mathbb{E}_{\mathbf{v} \sim \mathcal{D}_{\mathrm{Outlier}}}\Big[\mathrm{hinge}\big(\tau_{\mathrm{Outlier}} - E(\mathbf{v})\big)\Big], \qquad (17)$$

where $\mathcal{D}^t$ is the training set for task $t$, $\mathcal{H}$ is a cross-entropy or similar classification loss, $\widehat{\mathbf{y}}^{(\mathbf{x})}$ the model prediction, and $\mathbf{y}$ the ground-truth label for $\mathbf{x}$. The term $E(\mathbf{v})$ is an energy-based confidence measure, while penalizing overconfidence for outliers. Despite using only one prompt for all tasks, OVOR reports strong results by restricting overconfident predictions and minimizing overhead.

**LAE** reframes prompts as an instance of parameter-efficient fine-tuning (PEFT), incorporating additional modules (Adapters or LoRA) to mitigate forgetting [7]. It accumulates multi-task knowledge through online/offline PET modules and ensembles them at inference. Concretely, it defines an online PET $\boldsymbol{\theta}_{\mathrm{pet}}^{\mathrm{on}}$ for new tasks and an offline PET $\boldsymbol{\theta}_{\mathrm{pet}}^{\mathrm{off}}$ for older tasks:

$$\boldsymbol{\theta}_{\mathrm{pet}}^{\mathrm{off}} \; \leftarrow \; \alpha\, \boldsymbol{\theta}_{\mathrm{pet}}^{\mathrm{off}} \; + \; (1-\alpha)\, \boldsymbol{\theta}_{\mathrm{pet}}^{\mathrm{on}}, \qquad (18)$$

where $\alpha$ is close to 1 (EMA update). During inference, LAE combines the outputs of these two experts to better handle both novel and historical tasks.

### B.2 Hyperparameters and configurations

For all prompt-based CL methods, we employ the ADAM optimizer [15] with hyperparameters $\beta_1 = 0.990$ and $\beta_2 = 0.999$, following their original implementation. For prompt selection, we set the prompt pool size to 10 and the prompt length to 5, following the original implementations of L2P, DualPrompt, and HiDe-Prompt. For CODA-Prompt, we implement a cosine-decay learning rate strategy described in [31], while other prompt-based baselines use a fixed learning rate of $1.875 \times 10^{-3}$ by default.

We consistently use the mini-batch size of 16 for all methods to maintain a uniform computational load for each training iteration. For fair comparisons, each method performs 1875, 1080, and 2583 iterations per task for Split CIFAR-100 [18], Split ImageNet-R [12], and Split PlantDisease [25], respectively. This setup ensures that among prompt-based methods, iteration time itself correlates linearly with energy usage, as training wall-clock time similarly reflects energy consumption, as discussed in the main paper.

## C Competing methods with original setups

For the evaluation of baseline methods in the main paper, we adhere to the hyperparameters specified in the original studies, except for the *number of iterations per task* and *batch size*. We adopt a smaller batch size to account for on-device memory limitations, which prevent the use of the original batch sizes. Nonetheless, we present the results of REP-L2P, comparing with L2P, DualPrompt, CODA-Prompt, HiDe-Prompt, ConvPrompt, LAE, and OVOR using their original hyperparameters in Table 9 and Table 10, respectively. In these experiments, we measure wall-clock GPU time rather than iteration time to ensure a fair comparison across diverse original settings. The results demonstrate that although these methods are based on the ViT-B backbone, most of them consume significantly more memory than REP-L2P, primarily due to their originally large batch sizes.

Table 9: Results on Split CIFAR-100, with ViT-B as the backbone (following their original setups).

| Method | Acc. (↑) | Fgt. (↓) | GPU time (s) | Mem. (GB) |
|---|---|---|---|---|
| REP-L2P | 87.0±1.3 | 4.5±0.1 | 4,421 | 4.5 |
| L2P-B | 84.4±0.7 | 7.4±0.4 | 2,803 | 14.2 |
| DualPrompt-B | 85.3±0.8 | 5.2±0.1 | 2,624 | 12.3 |
| CODA-Prompt-B | 86.1±0.7 | 1.7±0.3 | 3,227 | 18.8 |
| HiDe-Prompt-B | 92.2±0.4 | 3.2±0.4 | 20,169 | 5.3 |
| ConvPrompt-B | 88.6±0.6 | 3.4±0.4 | 6,544 | 14.8 |
| LAE-Prefix10-B | 84.8±0.6 | 4.9±0.8 | 1125 | 2.8 |
| OVOR-B | 85.3±0.6 | 4.8±0.8 | 2,637 | 8.4 |

Table 10: Results on Split ImageNet-R, with ViT-B as the backbone (following their original setups).

| Method | Acc. (↑) | Fgt. (↓) | GPU time (s) | Mem. (GB) |
|---|---|---|---|---|
| REP-L2P | 75.3±1.5 | 3.6±0.3 | 2,542 | 4.5 |
| L2P-B | 59.7±0.8 | 9.7±0.5 | 1,610 | 14.2 |
| DualPrompt-B | 67.5±0.9 | 4.7±0.2 | 1,508 | 12.3 |
| CODA-Prompt-B | 70.5±0.7 | 1.6±0.1 | 1,854 | 18.8 |
| HiDe-Prompt-B | 75.0±0.5 | 2.4±0.6 | 11,588 | 5.3 |
| ConvPrompt-B | 76.3±0.5 | 3.6±0.5 | 3,760 | 14.8 |
| LAE-Prefix10-B | 70.8±0.8 | 8.8±0.9 | 646 | 14.8 |
| OVOR-B | 73.2±0.8 | 4.5±0.9 | 1,516 | 8.4 |

# D Preliminary empirical studies

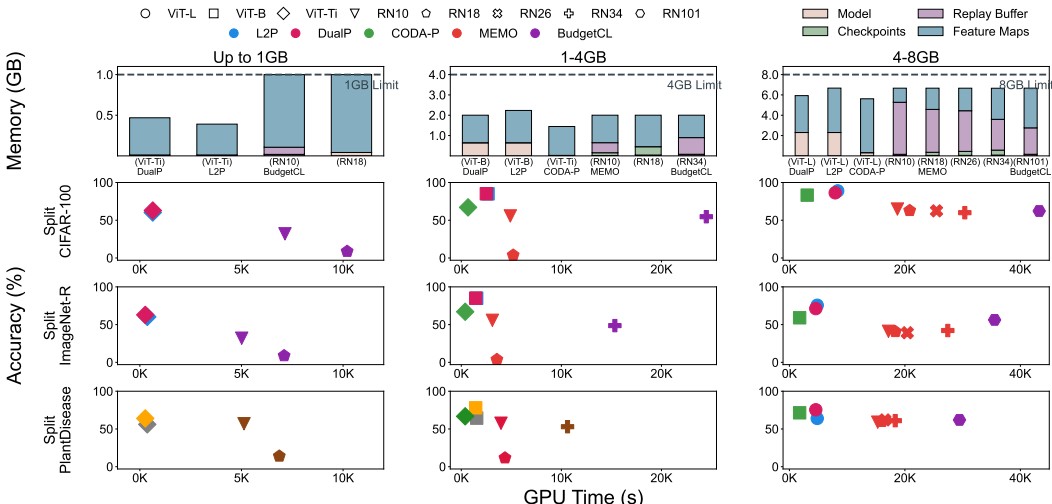

Figure 5: Cost-accuracy trade-offs of various ViT- and CNN-based methods over three different memory budgets: up to 1GB, 1–4GB, and 4–8GB. The memory breakdown of each method is in the first row. Experiments on Split CIFAR-100, Split ImageNet-R, and Split PlantDisease are on the second, third, and fourth row, respectively. ViT-based methods consistently outperform CNN-based methods by a large margin.

In this studies, we uncover the cost-accuracy trade-offs to highlight the impact of each baseline method on the accuracy gained relative to its training cost. Here, we limit the memory to a maximum of 8GB. We employ L2P [38], DualPrompt [37], and CODA-Prompt [31] as representative ViT-based prompting methods. To demonstrate the effectiveness of prompting methods, we also incorporate two state-of-the-art CNN-based methods, such as BudgetCL [27] and MEMO [43]. Both BudgetCL and MEMO improve model accuracy by leveraging spare memory to store and replay past samples, with MEMO additionally storing model checkpoints from history. These methods also use ImageNet

pre-trained models as backbones: ResNet-10 (RN10; 5M), ResNet-18(RN18; 11M), ResNet-26 (RN26; 14M), ResNet-34 (RN34;22M), and ResNet-101 (RN101; 43M) models.

For BudgetCL and MEMO, we adopt the SGD optimizer with a learning rate scheduling and a weight decay, following their original papers [27, 43]. Both methods construct mini-batches of 16 samples, with 8 samples from the new task and 8 from prior tasks in the replay buffer. We adopt the setup in [43] and train for 200 epochs across all datasets. Thus, BudgetCL and MEMO perform more iterations per epoch with a larger replay buffer, resulting in longer training time and higher energy usage. For fair comparisons, we group methods by memory usage.

Figure 5 visually represents the comparison results across device memory capacities and datasets. In each graph, the x-axis indicates GPU wall-clock time, which corresponds to energy cost (lower is better), while the y-axis indicates final average accuracy (higher is better). Thus, a more cost-effective method appears closer to the upper-left corner of the graph. Overall, ViT-based methods outperform CNN-based methods by a large margin, achieving 26–36% higher accuracy with 45–90% less time and energy spent under the same memory budget. We elaborate on two key characteristics that explain these results:

**Scaling across varying backbone networks.** As we can see in Fig. 5, ViT-based methods scale well, with larger backbone networks overall yielding higher accuracy. In contrast, CNN-based methods scale poorly with increased backbone size. Specifically, on Split ImageNet-R, a more challenging dataset, MEMO/RN34 is only 3.0% better than MEMO/RN18, despite being around twice larger. On Split CIFAR-100, increasing the backbone size yields only limited gains.

**Memory efficiency.** CNN-based methods are often considered more memory-efficient because they use backbones a magnitude smaller than ViT models. Thus, it is commonly believed that ample memory can be allocated to replay samples or the past model checkpoints [43], which help improve model performance. However, in CL, which involves typical training iterations, a substantial amount of memory must be allocated for feature maps [21], leaving little room for memory buffers. For instance, for a device memory range of 1–4GB in Table 11, the backbone of MEMO/RN18 consumes only 43MB of memory (6.5% of ViT-L), but feature maps occupy as much as 1.7GB (more than ViT-based methods). This causes MEMO/RN18 to be short on memory for the replay buffer, dramatically degrading model accuracy, as observed in Fig. 5.

Table 11: ViT- and CNN-based methods with specific backbone models are mapped to the memory capacity they can fit into. DualP is DualPrompt, and CODA-P is CODA-Prompt.

| Memory capacity | Edge device | ViT-based methods | CNN-based methods |
|---|---|---|---|
| 4–8GB | Jetson Xavier NX | L2P (ViT-L), DualP (ViT-L) CODA-P (ViT-B) | MEMO (RN26, RN34) BudgetCL (RN101) |
| 1–4GB | Jetson Nano | L2P (ViT-B), DualP (ViT-B) BudgetCL (RN34) | MEMO (RN10, RN18) |
| Up to 1GB | Raspberry Pi | L2P (ViT-Ti), DualP (ViT-Ti) | BudgetCL (RN10, RN18) |

# E  Extended ablation studies

As the paper mainly discusses the ablation study based on Split ImageNet-R, we here present an extended ablation studies of REP using Split CIFAR-100 (10 tasks). The results are shown in Table 12. Each row reflects performance with the corresponding component removed. Similar to the findings from Split ImageNet-R, the ablation of any component within REP for the Split CIFAR-100 dataset impacts resource efficiency. All components contribute to the reduction of computation time and memory usage, with AToM particularly standing out in both aspects of resource efficiency.

Table 12: Component ablation for Split CIFAR-100. Ablating any component of REP results in lower resource efficiency by increasing training time and memory consumption.

| Ablated components | Acc. (↑) | Fgt. (↓) | Iter. time (ms) | Mem. (GB) |
|---|---|---|---|---|
| REP-L2P | 87.0±1.3 | 4.5±0.1 | 240 | 4.5 |
| w/o (AToM + ALD) | 86.4±0.4 | 5.8±0.8 | 349 | 5.5 |
| w/o ($f_{efficient}$ + ALD) | 86.4±0.2 | 4.9±0.8 | 419 | 5.6 |
| w/o ($f_{efficient}$ + AToM) | 86.8±0.7 | 5.0±0.4 | 401 | 6.5 |
| w/o ALD | 85.9±0.5 | 5.0±0.7 | 270 | 4.5 |
| w/o AToM | 86.1±0.2 | 4.7±0.7 | 303 | 5.5 |
| w/o $f_{efficient}$ | 86.1±0.5 | 4.0±0.4 | 326 | 4.8 |

Next, we perform algorithm validation on Split CIFAR-100 (10 tasks), for which we present the results in Table 13. Consistent with the Split ImageNet-R results, using static token merging (ToMe) [2] or progressive layer dropping (PLD) [41] instead of adaptive token merging (AToM) or adaptive layer dropping (ALD) negatively affects model accuracy. This experiment further corroborates the efficacy of our proposed algorithms for optimizing resource usage without

Table 13: Algorithm validation for Split CIFAR-100. Changing our algorithms to conventional token merging (ToMe) or progressive layer dropping (PLD) harms model accuracy.

| | Acc. ($\uparrow$) | Fgt. ($\downarrow$) | Iter. time (ms) | Mem. (GB) |
|---|---|---|---|---|
| REP-L2P | 87.0±1.3 | 4.5±0.1 | 240 | 4.5 |
| w/ ToMe | 83.6±0.1 | 3.7±0.6 | 275 | 3.7 |
| w/ PLD | 84.4±0.4 | 4.9±0.8 | 259 | 4.5 |

compromising the model accuracy in CL setup. We further extend our analysis from Table 4 by evaluating two additional prompt-based CL methods, including HiDe-Prompt (denoted as REP-HiDeP) and ConvPrompt (denoted as REP-ConvP), with results detailed in Table 14. These results show a trend consistent with Table 4.

Table 14: REP over diverse # of merged tokens ($n$) and % of keep ratio ($\theta$), using HiDe-Prompt and ConvPrompt.

| Method | $n$ (w/ $\theta$=0.5) | Acc. ($\uparrow$) | Fgt. ($\downarrow$) | Iter. time (ms) | Mem. (GB) |
|---|---|---|---|---|---|
| REP-HiDeP | 2 | 78.6±1.5 | 2.0±0.3 | 251 | 7.3 |
| REP-HiDeP | 4 | 78.3±1.1 | 2.1±0.8 | 245 | 7.0 |
| REP-HiDeP | 8 | 78.0±1.2 | 2.0±0.9 | 227 | 6.3 |
| REP-ConvP | 2 | 79.1±1.0 | 3.4±0.5 | 460 | 5.5 |
| REP-ConvP | 4 | 78.8±1.3 | 3.5±0.6 | 450 | 5.1 |
| REP-ConvP | 8 | 78.5±0.8 | 3.7±1.1 | 417 | 4.1 |

| Method | $\theta$ (w/ $n$=8) | Acc. ($\uparrow$) | Fgt. ($\downarrow$) | Iter. time (ms) | Mem. (GB) |
|---|---|---|---|---|---|
| REP-HiDeP | 0.25 | 77.8±1.0 | 2.4±0.8 | 211 | 6.3 |
| REP-HiDeP | 0.50 | 78.0±1.2 | 2.0±0.9 | 227 | 6.3 |
| REP-HiDeP | 0.75 | 77.9±0.9 | 2.3±0.7 | 255 | 6.3 |
| REP-ConvP | 0.25 | 78.2±1.1 | 3.9±0.4 | 387 | 4.1 |
| REP-ConvP | 0.50 | 78.5±0.8 | 3.7±1.1 | 417 | 4.1 |
| REP-ConvP | 0.75 | 78.3±0.4 | 4.0±1.2 | 469 | 4.1 |

## F    Diverse task sequence lengths

We examine the scalability and robustness of our approach by employing two task sequence lengths, including 5 and 20 tasks, on the Split CIFAR-100 and Split ImageNet-R datasets, respectively. Longer task sequences simulate real-world scenarios more accurately, where CL methods require frequent model updates. The results are shown in Table 15 and Table 16, respectively. Across all scenarios, our method shows consistent preservation of both model accuracy and resource efficiency.

Table 15: Results on Split CIFAR-100 organized as 5 tasks (5T) and 20 tasks (20T), respectively.

| Method | Accuracy ($\uparrow$) | | | | Forgetting ($\downarrow$) | | | | Iter. time (s) | | Mem. (GB) | |
|---|---|---|---|---|---|---|---|---|---|---|---|---|
| | Split CIFAR-100 (5T) | | Split CIFAR-100 (20T) | | Split CIFAR-100 (5T) | | Split CIFAR-100 (20T) | | | | | |
| | w/o REP | w/ REP | w/o REP | w/ REP | w/o REP | w/ REP | w/o REP | w/ REP | w/o REP | w/ REP | w/o REP | w/ REP |
| L2P-L | 88.7±0.3 | 88.3±0.9 | 84.2±0.6 | 83.9±0.9 | 4.4±0.3 | 4.8±0.7 | 6.0±0.1 | 6.0±0.2 | 447 | 240 | 6.5 | 4.5 |
| DualPrompt-L | 87.9±0.5 | 87.2±0.9 | 83.7±0.7 | 83.0±0.9 | 4.4±0.2 | 4.7±0.8 | 6.3±0.6 | 6.8±0.9 | 424 | 208 | 5.9 | 4.3 |
| CODA-Prompt-L | 88.8±0.6 | 87.9±1.1 | 83.9±0.4 | 83.0±0.8 | 3.1±0.2 | 3.6±0.9 | 5.2±0.3 | 6.0±1.2 | 568 | 441 | 13.2 | 11.0 |
| HiDe-Prompt-L | 94.1±0.9 | 93.7±1.4 | 91.9±0.9 | 91.4±1.2 | 1.5±0.6 | 1.8±1.0 | 1.9±0.8 | 2.1±0.9 | 413 | 227 | 7.3 | 6.3 |
| ConvPrompt-L | 89.9±0.7 | 89.7±1.1 | 88.1±0.6 | 87.6±1.2 | 2.2±0.7 | 2.4±0.9 | 2.7±0.7 | 3.0±0.8 | 560 | 417 | 5.4 | 4.1 |
| LAE-Prefix10-L | 85.6±0.6 | 85.3±0.9 | 83.2±0.5 | 82.8±0.9 | 4.5±0.7 | 4.7±1.0 | 6.2±0.3 | 6.5±0.9 | 185 | 170 | 5.1 | 4.8 |
| OVOR-L | 86.9±0.8 | 86.4±1.1 | 85.4±0.6 | 85.0±0.9 | 3.2±0.7 | 3.6±1.0 | 3.0±0.4 | 3.2±0.9 | 467 | 362 | 7.5 | 6.8 |

## G    Additional datasets

We also expand our experiments on REP by incorporating an additional dataset: Split CUB-200 [32], which is designed for fine-grained image classification. Split CUB-200 contains 5,994 bird images

Table 16: Results on Split ImageNet-R organized as 5 tasks (5T) and 20 tasks (20T), respectively.

| Method | Accuracy (↑) | | | | Forgetting (↓) | | | | Iter. time (s) | | Mem. (GB) | |
|---|---|---|---|---|---|---|---|---|---|---|---|---|
| | Split ImageNet-R (5T) | | Split ImageNet-R (20T) | | Split ImageNet-R (5T) | | Split ImageNet-R (20T) | | | | | |
| | w/o REP | w/ REP | w/o REP | w/ REP | w/o REP | w/ REP | w/o REP | w/ REP | w/o REP | w/ REP | w/o REP | w/ REP |
| L2P-L | 77.5±0.4 | 77.1±0.8 | 72.8±0.2 | 72.6±0.6 | 1.0±0.2 | 2.0±0.7 | 4.7±0.5 | 4.5±0.5 | 447 | 240 | 6.5 | 4.5 |
| DualPrompt-L | 73.1±0.5 | 72.5±0.9 | 69.5±0.4 | 69.0±1.4 | 2.2±0.5 | 2.6±1.3 | 5.0±0.9 | 5.8±0.9 | 424 | 208 | 5.9 | 4.3 |
| CODA-Prompt-L | 76.8±0.6 | 76.0±1.1 | 73.4±0.2 | 72.7±1.1 | 1.1±0.1 | 1.7±0.9 | 5.2±0.3 | 5.9±0.9 | 568 | 441 | 13.2 | 11.0 |
| HiDe-Prompt-L | 79.1±0.5 | 78.6±1.2 | 78.0±0.6 | 77.5±1.3 | 1.7±0.5 | 2.1±0.9 | 2.3±0.6 | 2.6±1.1 | 413 | 227 | 7.3 | 6.3 |
| ConvPrompt-L | 79.9±0.8 | 79.5±1.0 | 78.3±0.5 | 77.8±1.1 | 2.7±0.6 | 3.0±0.9 | 2.9±0.9 | 3.3±0.9 | 560 | 417 | 5.4 | 4.1 |
| LAE-Prefix10-L | 71.7±0.8 | 71.4±1.3 | 69.8±0.7 | 69.3±0.9 | 4.1±0.3 | 4.3±0.9 | 5.0±0.4 | 5.4±0.9 | 185 | 170 | 5.1 | 4.8 |
| OVOR-L | 76.7±0.8 | 75.9±1.3 | 74.6±0.5 | 73.9±0.9 | 5.1±0.7 | 5.5±1.0 | 4.6±0.6 | 5.1±0.9 | 467 | 362 | 7.5 | 6.8 |

categorized into 200 classes, which are split into 5 tasks, each having 40 classes. As shown in Table 17, integrating REP achieves resource efficiency by bringing significant reductions in both training time and memory usage.

Table 17: Results on Split CUB-200, using ViT-L as the backbone model.

| Method | Acc. (↑) | | Fgt. (↓) | | Iter. time (ms) | | Mem. (GB) | |
|---|---|---|---|---|---|---|---|---|
| | w/o REP | w/ REP | w/o REP | w/ REP | w/o REP | w/ REP | w/o REP | w/ REP |
| L2P-L | 74.7±0.9 | 74.7±1.0 | 6.7±1.4 | 6.7±0.3 | 447(×1.9) | 240 | 6.5(×1.4) | 4.5 |
| DualPrompt-L | 72.4±0.9 | 72.1±0.9 | 7.8±0.3 | 7.9±0.8 | 424(×2.0) | 208 | 5.9(×1.4) | 4.3 |
| CODA-Prompt-L | 79.5±0.8 | 79.0±1.1 | 5.8±0.9 | 6.0±0.7 | 568(×1.3) | 441 | 13.2(×1.2) | 11.0 |
| HiDe-Prompt-L | 86.8±0.5 | 86.6±1.0 | 1.8±0.8 | 1.8±1.3 | 413(×1.8) | 227 | 7.3(×1.2) | 6.3 |
| ConvPrompt-L | 82.7±0.9 | 82.2±0.9 | 5.3±0.8 | 5.7±0.9 | 560(×1.3) | 417 | 5.4(×1.3) | 4.1 |
| LAE-Prefix10-L | 73.4±0.7 | 73.0±1.2 | 8.5±0.8 | 8.8±0.7 | 185(×1.1) | 170 | 5.1(×1.1) | 4.8 |
| OVOR-L | 78.6±0.7 | 78.0 | 5.9±0.9 | 6.3±0.9 | 467(×1.3) | 362 | 7.5(×1.1) | 6.8 |

# H Supplementary evaluations

**Self-supervised pre-training.** We evaluate REP with HiDe-Prompt [33] under two self-supervised pre-training paradigms, including iBOT21K [44] and DINO [3], using the ViT-B backbone on Split ImageNet-R (10 tasks). As shown in Table 18, REP reduces training time and memory usage by 14–18%, with only 0.6–0.8% of accuracy drop while forgetting remains improved.

**Comparison with analytic learning-based CL method.** We evaluate F-OAL [45] as the representative analytic learning-based CL method on Split CIFAR-100 (10 tasks). Table 19 demonstrates that REP-L2P can achieve compatible accuracy, with shorter training time. Since F-OAL is an online method, a perfect 1:1 comparison is tricky.

Table 18: Effect of REP with HiDe-Prompt under self-supervised pretraining paradigms.

| Method | Acc. (↑) | Fgt. (↓) | Iter. time (ms) | Mem. (GB) |
|---|---|---|---|---|
| HiDe-iBOT | 71.5±0.3 | 1.4±0.4 | 130 | 4.1 |
| REP-iBOT | 70.7±0.9 | 0.7±0.7 | 76 | 2.4 |
| HiDe-DINO | 68.6±0.2 | 1.8±0.2 | 130 | 4.1 |
| REP-DINO | 68.0±0.8 | 0.7±0.7 | 76 | 2.4 |

Table 19: Comparison of REP-L2P with F-OAL on Split CIFAR-100.

| Method | Acc. (↑) | Fgt. (↓) | Iter. time (ms) | Mem. (GB) |
|---|---|---|---|---|
| REP-L2P | 87.0±1.3 | 4.5±0.1 | 240 | 4.5 |
| F-OAL | 87.6±1.4 | 5.1±0.3 | 359 | 1.6 |

Nonetheless, these results suggest that REP can be competitive with F-OAL's energy efficiency while maintaining similar accuracy. Each method has distinct advantages and disadvantages depending on the context.

# I Extended main results including forgetting metric

We present the extended results in Table 1, including the forgetting metric. The result is shown in Table 20.

Table 20: Forgetting and computational cost of all competing methods on Split CIFAR-100, Split ImageNet-R, and Split PlantDisease datasets. We report forgetting, iteration time, and memory usage both without and with REP. Iteration time and memory usage are also shown as absolute values, with their multiples indicating how many times larger than REP.

| Model | Method | Forgetting (↓) | | | | | | Iter. time (ms) | | Mem. (GB) | |
| | | Split CIFAR-100 | | Split ImageNet-R | | Split PlantDisease | | | | | |
| | | w/o REP | w/ REP | w/o REP | w/ REP | w/o REP | w/ REP | w/o REP | w/ REP | w/o REP | w/ REP |
|---|---|---|---|---|---|---|---|---|---|---|---|
| ViT-L | L2P | 4.7±0.1 | 4.5±0.1 | 2.8±0.2 | 3.4±0.7 | 13.2±3.8 | 8.7±2.1 | 447(×1.9) | 240 | 6.5(×1.4) | 4.5 |
| | DualPrompt | 4.5±0.7 | 4.9±0.9 | 3.8±0.4 | 4.1±0.9 | 14.8±3.0 | 10.1±3.7 | 424(×2.0) | 208 | 5.9(×1.4) | 4.3 |
| | CODA-Prompt | 4.3±0.1 | 4.9±0.8 | 3.7±0.8 | 4.3±1.8 | 14.9±2.9 | 15.1±3.0 | 568(×1.3) | 441 | 13.2(×1.2) | 11.0 |
| | HiDe-Prompt | 1.7±0.6 | 1.8±1.1 | 2.0±0.5 | 2.0±0.9 | 1.0±0.8 | 1.1±1.3 | 413(×1.8) | 227 | 7.3(×1.2) | 6.3 |
| | ConvPrompt | 3.0±0.7 | 3.1±0.9 | 3.4±0.8 | 3.7±1.1 | 1.2±0.5 | 1.2±1.0 | 560(×1.3) | 417 | 5.4(×1.3) | 4.1 |
| | LAE-Prefix-10 | 5.4±0.8 | 5.5±1.0 | 4.5±0.5 | 4.6±1.2 | 15.5±0.8 | 15.9±1.4 | 185(×1.1) | 170 | 5.1(×1.1) | 4.8 |
| | OVOR | 2.9±0.8 | 3.1±1.4 | 4.6±0.9 | 4.9±1.8 | 14.5±2.5 | 14.9±3.1 | 467(×1.3) | 362 | 7.5(×1.1) | 6.8 |
| | UpperBound | – | – | – | – | – | – | 427 | – | 9.8 | – |
| ViT-B | L2P | 6.6±0.8 | 6.7±0.9 | 6.6±1.2 | 6.1±1.2 | 25.9±2.8 | 23.5±2.9 | 143(×1.4) | 102 | 2.3(×1.2) | 2.0 |
| | DualPrompt | 5.6±0.3 | 5.6±0.7 | 3.7±0.3 | 4.3±1.1 | 10.5±1.9 | 10.9±2.0 | 133(×1.3) | 101 | 2.1(×1.2) | 1.8 |
| | CODA-Prompt | 6.0±0.2 | 6.5±1.5 | 6.1±0.8 | 6.9±0.9 | 18.8±1.1 | 19.1±2.9 | 164(×1.1) | 151 | 6.9(×1.2) | 5.7 |
| | HiDe-Prompt | 1.6±0.8 | 1.6±1.4 | 1.2±0.8 | 1.9±1.3 | 1.0±0.8 | 1.2±1.0 | 130(×1.7) | 76 | 4.1(×1.7) | 2.4 |
| | ConvPrompt | 3.7±0.6 | 3.9±1.6 | 5.5±0.7 | 5.9±1.4 | 1.6±0.8 | 1.8±1.5 | 381(×1.4) | 279 | 2.2(×1.1) | 2.0 |
| | LAE-Prefix-10 | 6.8±0.9 | 5.9±0.7 | 9.3±0.9 | 9.2±1.0 | 16.9±0.8 | 16.9±1.2 | 57(×1.1) | 50 | 1.9(×1.1) | 1.8 |
| | OVOR | 5.7±0.6 | 5.9±1.5 | 4.9±0.7 | 5.5±1.4 | 20.8±0.9 | 21.4±1.6 | 134(×1.1) | 123 | 3.1(×1.1) | 3.0 |
| | UpperBound | – | – | – | – | – | – | 143 | – | 3.2 | – |
| ViT-Ti | L2P | 16.3±0.9 | 16.8±1.1 | 9.9±0.9 | 10.5±1.4 | 31.2±3.1 | 29.9±4.1 | 34(×1.1) | 30 | 0.5(×1.1) | 0.5 |
| | DualPrompt | 15.7±2.2 | 15.8±3.0 | 8.9±0.8 | 9.2±1.0 | 25.0±2.2 | 26.1±2.8 | 34(×1.1) | 31 | 0.4(×1.1) | 0.4 |
| | CODA-Prompt | 12.9±1.6 | 13.2±2.3 | 13.4±0.7 | 14.0±0.8 | 23.9±3.3 | 24.0±3.8 | 36(×1.2) | 30 | 2.8(×1.1) | 2.8 |
| | HiDe-Prompt | 6.8±0.7 | 6.6±1.3 | 2.2±0.5 | 1.9±1.3 | 1.0±0.8 | 1.0±0.4 | 23(×1.2) | 20 | 0.5(×1.1) | 0.4 |
| | ConvPrompt | 10.3±0.7 | 11.1±1.8 | 14.1±0.8 | 15.0±1.3 | 1.5±0.9 | 1.0±0.7 | 85(×1.9) | 45 | 0.5(×1.1) | 0.5 |
| | LAE-Prefix-10 | 17.1±0.3 | 17.1±0.9 | 17.0±0.5 | 17.8±0.9 | 27.6±0.8 | 27.9±1.4 | 15(×1.1) | 12 | 0.4(×1.1) | 0.4 |
| | OVOR | 12.0±1.5 | 12.9±2.7 | 12.8±0.8 | 13.1±1.5 | 19.7±3.1 | 18.8±3.9 | 35(×1.1) | 32 | 1.0(×1.1) | 0.9 |
| | UpperBound | – | – | – | – | – | – | 42 | – | 0.6 | – |

## J  Additional related works

### J.1  Online or few-shot CL

In on-device CL, the memory capacity for storing data from new tasks may be limited. Thus, one might consider adopting online or few-shot CL methods to save memory costs, as they require fewer new-task samples to be maintained in memory. Offline CL (our setting) typically yields better accuracy with higher resource usage. However, memory allocation for new samples is not a major constraint in offline CL. All prompt-based methods require 155–269MB to store new samples for both Split CIFAR-100 and Split ImageNet-R, accounting for only 3.4–5.9% of REP-L2P's memory usage. Moreover, modern DL frameworks can store data in storage and proactively prefetch upcoming mini-batches for training, virtually eliminating memory concerns for new samples.

### J.2  Advantages of REP over NAS

While NAS (Neural Architecture Search) can be integrated into REP to drop some layers, it usually involves searching the model from scratch, which is time- and computation-consuming. In contrast, our method applies adaptive schedules directly to an existing pre-trained model, significantly saving computation time. So, it can naturally adapt to device-internal resource conditions without relying on external server resources.

## K  Impact of varying hyperparameter values

Guided by the hyperparameter tuning strategies used in DualPrompt [37] and CODA-Prompt [31], we fine-tuned key hyperparameters for our ALD approach using REP-L2P. Our focus is primarily on optimizing the threshold parameter ($\alpha$) and the adjustment factor ($\tau$), which are crucial for ALD's adaptability and efficiency. We conducted a hyperparameter search using 5-fold cross-validation, for which we designate 20% of our training dataset as a validation set. Table 21 summarizes the results.

We further evaluate the impact of varying prompt-selection-related hyperparameters using REP-L2P on subsets of 5-datasets dataset introduced in [38]. Table 22 shows that the primary factor driving computation costs is prompt length, as it introduces more learnable parameters (e.g., with a pool size of 10, increasing prompt length from 5 to 10 increases memory usage by 500 MB and training time by 14%). In contrast, increasing the prompt pool size has almost no impact on training time and memory usage.

Table 21: Results of ALD's hyperparameter tuning via 5-fold cross-validation.

| $\alpha$ | $\tau$ | | | | | | | | | |
|---|---|---|---|---|---|---|---|---|---|---|
| | 6 | 8 | 10 | 12 | 14 | 15 | 16 | 17 | 18 | 20 |
| 0.6 | 74.32 | 74.62 | 74.75 | 74.45 | 74.87 | 74.99 | 74.93 | 75.12 | 74.82 | 74.82 |
| 0.7 | 74.17 | 74.38 | 74.45 | 74.98 | 74.18 | 75.06 | 75.15 | 75.26 | 74.43 | 74.31 |
| 0.8 | 74.48 | 74.50 | 74.41 | 74.36 | 74.87 | 75.29 | 74.91 | 75.20 | 74.99 | 74.39 |
| 0.9 | 74.49 | 74.36 | 74.77 | 74.38 | 74.58 | 75.33 | 75.34 | 75.42 | 74.88 | 74.96 |
| 0.95 | 74.45 | 74.76 | 74.39 | 74.54 | 74.69 | 75.39 | 75.32 | 75.38 | 74.76 | 75.13 |

Table 22: Effect of prompt pool size on subsets of 5-datasets. The prompt length is 10 by default.

| Pool size | Acc. ($\uparrow$) | Iter. time (ms) | Mem. (GB) |
|---|---|---|---|
| 10 | 62.73 | 240 | 4.5 |
| 15 | 63.07 | 240 | 4.5 |
| 20 | 63.40 | 240 | 4.5 |
| 30 | 63.78 | 241 | 4.5 |

## L   REP under varying computation budgets

We compare REP-L2P across various computation budgets defined by the number of iterations per task. The comparison results using Split CIFAR-100 (10 tasks) are shown in Table 23. Evidently, the accuracy of REP-L2P is comparable to that of L2P-L across six computation budgets, with the performance gap getting increasingly marginal at lower compute budgets. We observe that the overall accuracy for both methods does not drop significantly with reduced compute budgets. This is attributed to the fact that Split CIFAR-100 is considered a less complex CL benchmark.

Table 23: Impact of REP under various computation budgets (i.e., number of iterations per task).

| Method | 313 | 625 | 938 | 1250 | 1563 | 1875 |
|---|---|---|---|---|---|---|
| REP-L2P | 84.1 | 85.5 | 86.6 | 86.7 | 86.6 | 86.9 |
| L2P-L | 84.4 | 86.3 | 87.3 | 87.8 | 87.8 | 88.2 |

## M   REP on edge devices

Miro [22] introduces a dynamic approach for fine-tuning key design parameters of rehearsal-based CL methods to achieve high model accuracy while simultaneously reducing energy costs, i.e., high *cost-effectiveness*. Miro's methodology centers on identifying the suitable memory size within the device's capacity to accommodate both old and new samples for training. We compare REP-L2P with Miro directly on a reference edge device, NVIDIA Jetson TX2. This device is equipped with a 256-core NVIDIA Pascal GPU, 8GB of RAM, and a 32GB eMMC 5.1 drive. The RAM is a single unified memory shared by the CPU and GPU. We report energy usage as joules (J) obtained by multiplying power by time. Power usage is measured by reading the built-in sensor values provided by Jetson devices for the GPU, RAM, CPU, and I/O connection.

### M.1   Power usage breakdown

When implementing CL on edge devices, the majority of power consumption that influences energy usage during the training of a new task comes from GPU operations. Figure 6 shows power con-

sumption on NVIDIA Jetson TX2 across individual system components. Static power is measured when the system is inactive, while dynamic power is measured during the training of a ViT-B model. During the training process, power usage surges up to $6.5\times$, with the GPU contributing 60% to the dynamic power.

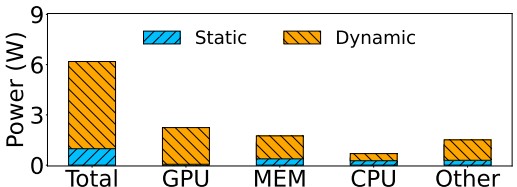

Figure 6: Power breakdown for training a ViT-B model on NVIDIA Jetson TX2.

## M.2   REP vs Miro

In this experiment, we explore energy-accuracy trade-offs of REP-L2P and Miro. Figure 7 visually represents the comparison results. In each graph, the x-axis signifies total energy usage (lower is better), while the y-axis signifies final average accuracy (higher is better). Hence, a more cost-effective strategy is positioned closer to the upper-left corner of the graph. Regarding Miro, we maintain the hyperparameters as suggested in the original work [22] but incorporate the use of pre-trained ResNet-18 (RN18; 11M), ResNet-34 (RN34; 22M), and ResNet-50 (RN50; 25M)[11] models instead of non-pre-trained ones to enhance overall performance. For REP-L2P, we vary the number of training iterations per task insertion to match the energy usage of Miro variants with different ResNet models.

When operating within the same energy budget, REP-L2P consistently outperforms Miro variants, achieving 22–33% higher accuracy across datasets. REP-L2P demonstrates superior cost-effectiveness, especially on Split ImageNet-R (10 tasks) compared to Split CIFAR-100 (10 tasks). Both REP-L2P and Miro prove to be memory-efficient to some extent as they fit comfortably within the on-device memory capacity. Although we explore larger ResNet models for Miro, we do not observe the accuracy levels comparable to REP-L2P for either dataset. This experiment also underscores the importance of specifically optimizing vision transformers for on-device CL scenarios to advance performance boundaries.

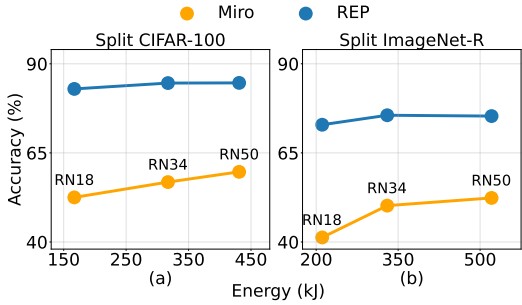

Figure 7: Energy-accuracy trade-offs between REP-L2P and Miro on Split CIFAR-100 (10 tasks) and Split ImageNet-R (10 tasks) . To provide a spectrum of energy-accuracy trade-offs, we use pre-trained ResNet-18 (RN18), ResNet-34 (RN34), and ResNet-50 (RN50) for Miro.

## N   Societal impacts

The proposed method widely promotes AI for robotics agents and surveillance systems with high efficiency. Once the edge AI is easily deployable by the proposed method, the system may be used for monitoring unwanted mass populations. Then, private information such as identity, clothing information, and personal attributes (e.g., age, gender, etc.) could be obtained by adversaries. Although our method has no intention to foster such problematic cases, we do not currently propose secure solutions to prevent them from happening.

