# OpenReview forum: "REP: Resource-Efficient Prompting for Rehearsal-Free Continual Learning"
_NeurIPS.cc/2025/Conference — NeurIPS 2025 poster_

### Official Review · Reviewer_W2oY · 2025-06-30

**Clarity:** 3
**Significance:** 2
**Originality:** 3
**Rating:** 4
**Confidence:** 5

**Summary:**

The paper introduces a novel strategy that reduces training time and computational memory costs in rehearsal-free, prompt-based continual learning setup, to facilitate real-world scenario deployment. The proposed method primarily optimizes two stages: prompt selection and prompt update. To improve in both stages, the paper proposes three techniques which trade a slight accuracy drop for improved resource efficiency. Specifically, the authors introduce a lightweight model with reduced depth and width for prompt selection. This technique also requires a non-trainable random projection to map back to original dimensions. Empirical results shows that this technique preserves 97% of representational similarity compared to a larger model.

The two other main techniques introduced, Adaptive Token Merging (AToM) and Adaptive Layer Dropping (ALD), are based on the analysis of which backbone blocks contribute most to the final loss. By measuring the mean attention distance they found that shallower layers capture more meaningful representations of the input. These findings are incorporated into AToM, which employs a scheduler that dynamically adjust the number of tokens to merge based on layer depth. Merging occurs more aggressively in deeper layers than in shallow layers, preserving important task-related information. ADL, on the other hand, is regulated by a layer-keeping probability that is also influenced by the degree of token merging performed by AToM in each layer. Since deeper layers tend to merge more tokens, they are more likely to be dropped.

The experiment results indicate that with by accepting a small trade-off in accuracy performance, the method achieves faster training and reduced memory usage.

**Questions:**

- Could the authors provide a more detailed, quantitative exploration of this trade-off? Specifically, can they present tables or graphs that clearly map different levels of resource saving to their corresponding accuracy degradation on the tested datasets?


- Do they have insights into how their current prompt selection mechanism would perform when encountering truly novel tasks or highly divergent domains that might not align well with the existing prompts?

**Ethical Concerns:**

["NO or VERY MINOR ethics concerns only"]

**Final Justification:**

After reading the rebuttal, the additional experiments, and the discussion with the AC and other reviewers, I find that the paper makes a practical, well-argued contribution to resource-efficient rehearsal-free prompt CL. While some concerns remain, the overall evidence supports the claim of meaningful efficiency gains with small accuracy trade-offs. On balance, reasons to accept outweigh the limitations, so I keep a borderline-accept recommendation.

**Limitations:**

The stated "curriculum limitation" is a technical detail that requires further clarification. The authors should elaborate on its meaning and explain why the inability to determine the learning curriculum constitutes a significant limitation. Furthermore, it is crucial to provide more granular and practical guidance on the accuracy-resource trade-off. Similar to the curriculum learning constraint, the paper should investigate how the method performs when truly novel and unexpected tasks or domains are introduced.

**Paper Formatting Concerns:**

I did not notice any major formatting issues.

**Quality:**

2

**Strengths And Weaknesses:**

The paper introduces an innovative strategy designed to reduce computational costs in continual learning, thereby enhancing the applicability of these systems in real-world scenarios, particularly in resource-constrained environments. This is achieved through three well-conceived techniques: a lightweight model with random projection, Adaptive Token Merging (AToM), and Adaptive Layer Dropping (ALD). These techniques operate synergistically to optimize both the prompt selection and prompt update stages.

While a trade-off analysis is included, understanding its granularity is crucial to allow users to effectively navigate this trade-off based on their specific requirements. Although the composition of the prompt pool has been analyzed, prompt selection relies on identifying the prompt with the highest similarity to a given input. This approach might lead to only a subset of prompts being frequently chosen and subsequently updated, perhaps because they are better suited to different input types. This behavior is partly justified by findings that increasing the overall prompt pool size does not yield significant improvements. However, this selection mechanism could become a limitation when dealing with data originating from highly diverse domains.

---

> ### Author Rebuttal · Authors · 2025-07-31
>
> Thank you for your review and constructive comments.
>
> > ‘Q1: Could the authors provide a more detailed, quantitative exploration of this trade-off? Specifically, can they present tables or graphs that clearly map different levels of resource saving to their corresponding accuracy degradation on the tested datasets?’
>
> **[Response]**
> Thank you. The tables below extend Table 4 of our main paper, showing how varying the intensity of AToM and ALD affects resource savings alongside accuracy drop and forgetting increase. REP provides flexible resource savings under different computational budgets while incurring only moderate accuracy loss. Nonetheless, there is a clear trade-off: more aggressive computation skipping leads to greater accuracy degradation. We will include these results in the final version.
>
> | N (w/ θ = 0.5) | Acc | GPU Time (s) | Memory (GB) |
> |---------------:|---------:|-------------:|------------:|
> | Baseline | 75.6 | 4735.2 | 6.5 |
> | 1 | 75.5 (-0.1 %) | 2833.6 (-40.1 %) | 5.5 (-15.4 %) |
> | 2 | 75.2 (-0.4 %) | 2808.4 (-40.7 %) | 5.5 (-15.4 %) |
> | 4 | 75.3 (-0.3 %) | 2714.0 (-42.7 %) | 5.2 (-20.0 %) |
> | 6 | 75.1 (-0.5 %) | 2667.5 (-43.7 %) | 4.8 (-26.2 %) |
> | 8 | 75.3 (-0.3 %) | 2542.4 (-46.3 %) | 4.5 (-30.8 %) |
> | 10 | 73.6 (-2.0 %) | 2413.8 (-49.0 %) | 4.1 (-36.9 %) |
>
> | θ (w/ n = 8) | Acc | GPU Time (s) | Memory (GB) |
> |-------------:|---------:|-------------:|------------:|
> | Baseline | 75.6 | 4735.2 | 6.5 |
> | 0.9 | 74.3 (-1.3 %) | 2987.8 (-38.1 %) | 4.5 (-30.8 %) |
> | 0.8 | 74.2 (-1.4 %) | 2935.5 (-38.2 %) | 4.5 (-30.8 %) |
> | 0.7 | 74.4 (-1.2 %) | 2861.4 (-39.6 %) | 4.5 (-30.8 %) |
> | 0.6 | 74.8 (-0.8 %) | 2705.6 (-42.9 %) | 4.5 (-30.8 %) |
> | 0.5 | 75.3 (-0.3 %) | 2542.4 (-46.3 %) | 4.5 (-30.8 %) |
> | 0.4 | 74.7 (-0.9 %) | 2489.8 (-47.4 %) | 4.5 (-30.8 %) |
> | 0.3 | 74.4 (-1.2 %) | 2413.3 (-49.0 %) | 4.5 (-30.8 %) |
> | 0.2 | 73.2 (-2.4 %) | 2362.4 (-50.1 %) | 4.5 (-30.8 %) |
> | 0.1 | 72.9 (-2.7 %) | 2301.7 (-51.4 %) | 4.5 (-30.8 %) |
>
>
>
>
>
>
>
>
>
>
>
>
> > ‘Q2: The stated "curriculum limitation" is a technical detail that requires further clarification. The authors should elaborate on its meaning and explain why the inability to determine the learning curriculum constitutes a significant limitation.’
>
> **[Response]**
> Thank you for asking for clarification. In Section 5, we intended to clarify the scope of our work: we focus on a standard continual learning setting where task arrivals are predefined as part of the problem formulation. In this setting, REP is designed to operate under the given task order.
>
> Curriculum learning, which explicitly determines or optimizes the task sequence, addresses a related but distinct problem. Since our study does not include curriculum-based task scheduling, REP has not been evaluated in scenarios where the task order can be deliberately arranged. Investigating how REP could be combined with curriculum learning strategies is an interesting direction for future research.
>
> > ‘Q3: Do they have insights into how their current prompt selection mechanism would perform when encountering truly novel tasks or highly divergent domains that might not align well with the existing prompts?’
>
> **[Response]**
> Thank you for the comment. To clarify, the similarity-based prompt selection mechanism is part of the baseline CL methods rather than REP itself. To simulate more diverse domains, we conducted preliminary experiments with subsets of the 5-datasets benchmark, introduced in [3]. The results are reported below:
>
>
>
> | Pool size  | Acc | Memory |
> |-----------------|-------|-------|
>  | 10 | 62.73 | 4.95 |
>  | 15 | 63.07 | 4.95 |
>  | 20 | 63.40 | 4.95 |
>  | 30 | 63.78 | 4.95 |
>
> The average accuracy is lower due to the stronger domain drifts, and the optimal pool size shifted to larger values. Given the results, we agree that for tasks that cover more diverse domains, similarity-based prompt-selection mechanisms can become a limitation, as also discussed in L2P [3]. When testing L2P on a collection of 5 commonly used datasets, increasing the pool size leads to proportional improvements in accuracy.
> Nonetheless, as shown in the table above, the pool size is not a dominant factor for memory usage or computation time. For tasks that contain higher drifts or cover more domains, we can adopt a larger pool size without compromising the cost-efficiency that REP brings.
>
> If we have misunderstood the reviewer’s comment, please advise us.
>
> Reference
>
> [2] Koh et al, WILDS: a benchmark of in-the-wild distribution shifts, ICML 2021.
>
> [3] Wang et al, Learning to Prompt for Continual Learning, CVPR 2022.

---

> > ### Author Response · Authors · 2025-08-04
> > **Dear Reviewer**
> >
> > Dear reviewer W2oY,
> >
> > We sincerely appreciate your thoughtful review and valuable feedback, which have been invaluable in improving our research. It has been a while since we posted our last response, so we would like to follow up to see whether our response has sufficiently addressed your concerns. If there is anything else the reviewer wants us to address more, please let us know. We would be happy to engage in any further discussion.
> >
> > Best regards,
> >
> > Authors

---

> ### Comment · Area_Chair_RW1S · 2025-08-05
> **post-rebuttal comments**
>
> Dear reviewer W2oY,
>
> As you may have seen already, the authors have responded to the questions in your initial review. Can you please share your thoughts post rebuttal, once you've had a chance to see the author responses and also the other reviews?
>
> Best, AC

---

> > ### Comment · Reviewer_W2oY · 2025-08-05
> >
> > Thanks for the answers and the additional experiments.
> >
> > - The authors' response, which includes extended experiments to simulate diverse domains, indicates that a larger prompt pool can enhance average accuracy. They also report that this increase does not impact memory consumption. Furthermore, the authors state that in certain scenarios, "we can adopt a larger pool size without compromising the cost-efficiency that REP brings." This assertion appears to contradict the justification for their hyperparameter choices in Appendix L, where they state, "when we increase the pool size and prompt length, there is a notable increase in computation cost." The authors should clarify this apparent discrepancy.
> >
> > - In response to Question 3 from reviewer tV8e, the authors provided a more in-depth analysis of how REP can increase accuracy, with additional experiments conducted on the Split ImageNet-R dataset divided into 10 tasks. However, the reviewer's original question pertained to the Split PlantDisease dataset, as shown in Table 1. Could the authors elaborate on why they selected ImageNet-R instead and explain what makes it the more appropriate or informative benchmark in this context?
> >
> > - The paper states that the Split ImageNet-R dataset is divided into 20 tasks. For the new experiments, why did the authors use a 10-task configuration instead of the 20-task setup used in the other experiments? Maintaining a consistent experimental setup would be beneficial for the sake of comparison.

---

> ### Author Response · Authors · 2025-08-06
>
> Dear reviewer W2oY,
>
> Thank you for your thoughtful follow-up questions. We address each point below.
>
> > ‘Q1: Discrepancy between "no mem increase" and Appendix L’s "computation cost increase’
>
> **[Response]**
> We apologize for the confusion caused by the statement in Appendix L (Figure 9). The primary factor driving computational costs is prompt length, not pool size. Increasing the prompt length (i.e., the number of trainable prompts per task) introduces more parameters to learn, which in turn increases both memory usage and computation time. In contrast, enlarging the prompt pool mainly requires additional memory to store candidate prompts; this cost is negligible compared with the backbone parameters and activations. Selecting prompts from the pool is essentially a lightweight lookup and has almost no impact on iteration time. To clarify this, we have updated the table to show training GPU wall-clock time.
>
> | Pool Size  | Acc | GPU Time (s) | Memory (GB) |
> |-----------------|-------|-------|-------|
>  | 10 | 62.73 | 2542.4 | 4.95 |
>  | 15 | 63.07 | 2544.1 | 4.95 |
>  | 20 | 63.40 | 2545.6 | 4.95 |
>  | 30 | 63.78 | 2548.3 | 4.95 |
>
> We also measured how prompt length affects resource usage to support our claim. With a pool size of 10, increasing the prompt length from 5 to 10 adds roughly 500MB of memory usage and increases GPU Time by about 14% (from 2542.4s to 2896.1s). We will revise the caption and text for Figure 9 to emphasize this point and eliminate confusion.
>
> > ‘Q2: Why Split ImageNet-R instead of Split PlantDisease’
>
> **[Response]**
> We apologize for misjudging the relevance of Split PlantDisease to reviewer tV8e’s Q3. Thank you for bringing this to our attention. REP is designed to reduce computation cost and memory usage with minimal loss of accuracy. The accuracy gains observed in some cases of Split PlantDisease (e.g., L2P with ViT-L) and Split ImageNet-R (e.g., HiDe-Prompt with ViT-Ti) were not anticipated by design. Nonetheless, as suggested by reviewer tV8e, we conducted a deeper analysis to better understand these gains, assuming they might be related to reduced variance in token norms (a proxy of noisy features). We have updated the original table to include results from L2P with ViT-L on the Split PlantDisease dataset.
>
> | Dataset | Method | Acc | Fgt | Norm. Variance |
> |------------|------------|------------|------------|------------|
> | Split ImageNet-R | L2P-L | 75.6±1.0 | 2.8±0.2 | 0.29 |
> | Split ImageNet-R | REP-L2P | 75.3±1.5 | 3.4±0.7 | 0.20 (-30%) |
> | Split PlantDisease | L2P-L | 75.9±3.1 | 13.2±3.8 | 0.32 |
> | Split PlantDisease | REP-L2P | 81.1±3.9 |8.7±2.1 | 0.21 (-34%) |
>
> REP reduces normalized activation variances on both PlantDisease and ImageNet-R. However, this reduction in variance does not translate into improved accuracy for Split ImageNet-R, suggesting that variance reduction alone does not fully explain the accuracy gains. We agree with reviewer tV8e that this is an intriguing phenomenon that warrants further investigation and, with the reviewers’ approval, we propose to continue exploring the underlying causes through more extensive studies.
>
> > ‘Q3: Why 10-task instead of 20-task for ImageNet-R’
>
> **[Response]**
> L234-235 of our main paper specifies that ImageNet-R is divided into 10 tasks, each containing 20 classes. We have used this setup consistently across all evaluations in the main paper and supplementary experiments. Only in Appendix C do we alter the number of tasks---Split CIFAR-100 and Split ImageNet-R into 20 tasks and 5 tasks---to study REP under different task sequence lengths. For your reference, we copy the 20-task results of REP-L2P from Table 10 and 11 in Appendix C.
>
> | Dataset | Method | Acc | Fgt | Iter. Time (ms) | Memory (GB) |
> |------------|------------|------------|------------|------------|------------|
> | Split CIFAR-100 | L2P-L | 84.2±0.6 | 6.0±0.1 | 447 | 6.5 |
> | Split CIFAR-100 | REP-L2P | 83.9±0.9 | 6.0±0.2 | 240 (-46%) | 4.5 (-31%) |
> | Split ImageNet-R | L2P-L | 72.8±0.2 | 4.7±0.5 | 447 | 6.5 |
> | Split ImageNet-R | REP-L2P | 72.6±0.6 | 4.5±0.5 | 240 (-46%) | 4.5 (-31%) |

---

> > ### Comment · Reviewer_W2oY · 2025-08-07
> >
> > Thank you to the authors for their response and clarifications. After carefully reading the rebuttal and the other reviews, I will keep my initial rating.

---

### Official Review · Reviewer_jSzb · 2025-07-02

**Clarity:** 2
**Significance:** 2
**Originality:** 2
**Rating:** 3
**Confidence:** 4

**Summary:**

This paper proposed a EFCIL method that based on prompt tuning. REP consists of two parts: The Adaptive Token Merging AToM and Adaptive Layer Dropping ALD.
The work first show that ViT behave in away that earlier layer focus on more local features while deeper layers focus on global information by mean attention distance analysis.
AToM dynamically merge tokens based on task specific information and layer specific information to allow more aggressive merging in deeper layers. To improve computation efficiency, AToM relies on a smaller model i.e ViT tiny to compute query feature for prompt selection in large model i.e ViT Base
ALD meanwhile adaptively drop information in ViT layed based on number of merged tokens in AToM. This allows more aggressive dropping in later layers as earlier layer contains richer features information.

**Questions:**

what is the baseline method  used in Table 4? Is it possible to show graph of ACC and forgetting of different n and theta for different CL method when combined with REP?
For figure 4. What happen if Random drop have different aggressiveness? Will it have a more closer performance compared to ALD?
For AToM, as shown in Figure 3, gradient w.r.t prompt is significant and contribute essential part to CF? Is it possible to improve AToM by involving Fisher Information Matrix like in EwC to have a gradient informed prompt merging?

**Ethical Concerns:**

["NO or VERY MINOR ethics concerns only"]

**Final Justification:**

Dear Authors, thanks for your response and updates. It's not fully clear if the trade-off between performance drop and computational gain is worthwhile across all settings. so I will keep my score as borderline reject.

**Limitations:**

The authors have thoroughly discussed limitations of REP in Section 5.

**Paper Formatting Concerns:**

No formatting issue

**Quality:**

2

**Strengths And Weaknesses:**

Strength
This work show good motivational experiment by showing ViT layers behaviour in mean attention distances which justify the need of more aggressive AToM and ALD. The ablation studies and comparison to other dropping strategies also further solidifies the claim as ALD perform well. Proposed method also achieve at least 10% faster computational time and 10% less memory consumption. The proposed method is tested in large number of different CL methods which proven it's adaptability and modularity. This work shows that it is possible to use lightweight backbone to perform approximated query selection on huge model, which is potentially beneficial for future works.
Weakness
REP performance in average accuracy drops compared to vanilla methods. Only very small amount of cases where REP actually help the AA. This raises the question on whether the computational speed and GPU memory saving is significant enough to justify the drop in performance.

---

> ### Author Rebuttal · Authors · 2025-07-31
>
> Thank you for your detailed review and highlighting both the strengths of REP and areas for improvement.
> > This raises the question on whether the computational speed and GPU memory saving is significant enough to justify the drop in performance.
>
>
> **[Response]**
> We acknowledge that in specific setups (e.g., CODA-Prompt), REP might cause a 1–2% accuracy drop in exchange for noticeable efficiency gains. In many other cases, however, the drop is negligible; for instance, in Split PlantDisease, REP even outperformed the baseline. This shows that the performance impact varies across tasks and baseline methods. Importantly, in edge-centric contexts, a small accuracy gap is often outweighed by the substantial energy and memory benefits, making the trade-off justifiable.
>
> > ‘Q1: What is the baseline method used in Table 4? Is it possible to show graph of ACC and forgetting of different n and theta for different CL method when combined with REP?’
>
> **[Response]**
> In the main paper, our ablation study primarily uses L2P without REP as the baseline CL method on Split ImageNet-R with a ViT-L backbone (additional results on Split CIFAR-100 are provided in Appendix H). Below, we present extended results for Table 4, including other CL methods such as HiDe-Prompt (‘HiDeP’) and ConvPrompt (‘ConvP’). These results demonstrate REP’s effectiveness across diverse CL methods: accuracy drops by less than 1% for moderate settings (n=4-8, θ=0.5-0.75), while forgetting remains stable or improves thanks to adaptive merging and dropping that preserve task-specific features. See Section 4.3 of the main paper for further studies, including non-prompting and self-supervised CL methods for REP’s generalizability.
>
> | Method | n (w/ theta=0.5) | Acc | Fgt | GPU Time (s) | Mem (GB) |
> |------------|------------|------------|------------|------------|------------|
> | HiDeP | 2 | 78.6±1.5 | 2.0±0.3 | 2654.8 | 7.3 |
> | HiDeP | 4 | 78.3±1.1 | 2.1±0.8 | 2597.1 | 7.0 |
> | HiDeP | 8 | 78.0±1.2 | 2.0±0.9 | 2404.7 | 6.3 |
> | ConvP | 2 | 79.1±1.0 | 3.4±0.5 | 4876.8 | 5.5 |
> | ConvP | 4 | 78.8±1.3 | 3.5±0.6 | 4770.8 | 5.1 |
> | ConvP | 8 | 78.5±0.8 | 3.7±1.1 | 4417.4 | 4.1 |
>
> | Method | theta (w/ n=8) | Acc | Fgt | GPU Time (s) | Mem (GB) |
> |------------|------------|------------|------------|------------|------------|
> | HiDeP | 0.25 | 77.8±1.0 | 2.4±0.8 | 2234.0 | 6.3 |
> | HiDeP | 0.5 | 78.0±1.2 | 2.0±0.9 | 2404.7 | 6.3 |
> | HiDeP | 0.75 | 77.9±0.9 | 2.3±0.7 | 2705.3 | 6.3 |
> | ConvP | 0.25 | 78.2±1.1 | 3.9±0.4 | 4103.8 | 4.1 |
> | ConvP | 0.5 | 78.5±0.8 | 3.7±1.1 | 4417.4 | 4.1 |
> | ConvP | 0.75 | 78.3±0.4 | 4.0±1.2 | 4969.6 | 4.1 |
>
> These results reveal REP's efficacy across diverse CL methods: Acc drops <1% for moderate n=4-8/theta=0.5-0.75, with Fgt stable or improved due to adaptive merging/dropping preserving task features. (Also, please refer to section 4.3 in our main paper for additional study, including non-prompting CL methods and self-supervised CL methods for REP’s generalizability.)
>
>
> > ‘Q2: For figure 4. What happen if Random drop have different aggressiveness? Will it have a more closer performance compared to ALD?’
>
> **[Response]**
> In Figure 4, Random Drop showed lower performance than ALD with a 25% drop. We further examined how varying aggressiveness (drop rate) affects the trade-off between accuracy and GPU time: We experimented with drop rates of 20%, 25%, 30%, and 50% on Split ImageNet-R using the ViT-L backbone, summarized in the table below. The results show that increasing the aggressiveness of Random Drop (30–50%) reduces GPU time below ALD but causes a significant accuracy drop (65–71%), making it less practical for CL. In contrast, ALD adaptively adjusts the drop ratio, maintaining resource efficiency with minimal accuracy loss.
>
> | Method   | Acc | GPU Time (s) |
> |-----------------|-------|-------|
>  | ALD | 75.8±0.5 | 4130 |
>  | Random-20% | 71.7±0.4 |4209 |
>  | Random-25% | 70.6±0.7 | 4014 |
>  | Random-30% | 68.8±0.6 | 3964 |
>  | Random-50% | 65.4±0.8 | 3743 |
>
> > ‘Q3: For AToM, as shown in Figure 3, gradient w.r.t prompt is significant and contribute essential part to CF? Is it possible to improve AToM by involving Fisher Information Matrix like in EwC to have a gradient informed prompt merging?’
>
> **[Response]**
> Yes, Fig. 3 shows that prompt gradients play a key role in mitigating CF, which motivates AToM’s prompt-preserving merging strategy. Incorporating the Fisher Information Matrix (FIM) for gradient-informed merging is indeed an interesting idea. We tested this approach using L2P-L (ViT-L backbone) on Split ImageNet-R. The performance improvement was limited (+0.1–0.2% in accuracy), while GPU time and memory usage increased by about 10% (e.g., baseline REP GPU time: 2542.4s → 2796.6s, memory: 4.5GB → 4.95GB). The additional computational cost of FIM reduced overall resource efficiency.
>
> We do not consider this a fundamental limitation but note that it is nontrivial to achieve practical gains. With sufficient optimization, gradient-informed merging remains a promising direction for future work.

---

> > ### Author Response · Authors · 2025-08-04
> > **Dear Reviewer**
> >
> > Dear reviewer jSzb,
> >
> > We sincerely appreciate your valuable feedback and constructive comments. Given that some time has passed since we shared our response, we kindly ask whether it has adequately addressed your concerns. Please let us know if there is anything else we need to address. We would be happy to discuss further. Otherwise, if you find our response reasonably satisfactory, we would greatly appreciate it if you could consider re-evaluating your initial rating.
> >
> > Best wishes,
> >
> > The authors.

---

> ### Comment · Area_Chair_RW1S · 2025-08-05
> **post-rebuttal comments**
>
> Dear reviewer jSzb,
>
> As you may have seen already, the authors have responded to the questions in your initial review. Can you please share your thoughts post rebuttal, once you've had a chance to see the author responses and also the other reviews?
>
> Best, AC

---

### Official Review · Reviewer_tV8e · 2025-07-02

**Clarity:** 3
**Significance:** 3
**Originality:** 3
**Rating:** 4
**Confidence:** 5

**Summary:**

This paper introduces REP (Resource-Efficient Prompting), a framework designed to improve the computational and memory efficiency of prompt-based, rehearsal-free continual learning (CL) methods for vision tasks. The authors propose three complementary techniques: (1) a lightweight surrogate model for swift prompt selection, (2) Adaptive Token Merging (ATOM) to intelligently reduce the number of input tokens while preserving task-specific prompt information, and (3) Adaptive Layer Dropping (ALD), which dynamically skips transformer layers based on the token merging ratio from ATOM to reduce computational load. Through extensive experiments on various datasets and Vision Transformer (ViT) backbones, the paper demonstrates that REP can significantly reduce iteration time and memory usage (by up to 51% and 48% respectively) with only a minimal trade-off in model accuracy.

**Questions:**

Please address the concerns raised in the "Weaknesses" section.

**Ethical Concerns:**

["NO or VERY MINOR ethics concerns only"]

**Final Justification:**

I appreciate the authors’ thorough rebuttal and find that they have satisfactorily addressed all of my concerns. I therefore uphold my original positive evaluation.

**Limitations:**

yes

**Quality:**

3

**Strengths And Weaknesses:**

Strengths:
1. The paper addresses the critical and highly relevant challenge of deploying continual learning models on resource-constrained edge devices.
2. The empirical evaluation is exceptionally thorough. The authors test REP across three datasets, three ViT backbone sizes, and integrate it into seven different state-of-the-art prompt-based CL methods.
3. The framework delivers substantial improvements in resource efficiency for a negligible drop in accuracy.

Weaknesses:
1. The core technical components, while cleverly adapted, are not fundamentally new concepts. The methodology relies on adaptive versions of existing techniques like Token Merging and Progressive Layer Dropping. While the paper's contribution lies in the intelligent, adaptive scheduling and combination of these techniques for the CL context, the work could be perceived as more of an engineering or systems contribution rather than a fundamental algorithmic breakthrough.
2. The proposed methods, particularly ALD, introduce new hyperparameters such as the merging threshold τ and the adjustment factor α. While the appendix provides a hyperparameter search, the paper lacks clear intuition or guidelines on how to set these parameters for new datasets or model architectures without performing expensive, task-specific tuning. This could potentially limit the "plug-and-play" applicability of the framework.
3. The paper notes that in some cases (e.g., L2P with ViT-L), REP can even improve model accuracy. The authors hypothesize that this might be because ALD and ATOM help eliminate redundant computations, effectively acting as a form of regularization. This is a very interesting phenomenon, but the paper fails to provide a deeper analysis to support this hypothesis. It is unclear whether this effect is due to the suppression of noisy features or some other mechanism. Adding some visualizations or quantitative analysis to investigate this would have significantly enhanced the paper's depth.

---

> ### Author Rebuttal · Authors · 2025-07-31
>
> Thank you for your detailed review and constructive comments. Here we address your initial concerns below.
>
> > ‘Q1: The core technical components, while cleverly adapted, are not fundamentally new concepts. The methodology relies on adaptive versions of existing techniques like Token Merging and Progressive Layer Dropping. While the paper's contribution lies in the intelligent, adaptive scheduling and combination of these techniques for the CL context…’
>
> **[Response]**
> We appreciate the reviewer's observation that REP incorporates ideas related to Token Merging and Progressive Layer Dropping. However, REP introduces new algorithmic elements that go beyond simply reusing these techniques: REP introduces adaptive mechanisms for both token merging and layer dropping that are specifically designed for rehearsal-free continual learning. AToM selectively excludes prompt tokens and progressively merges tokens more aggressively in deeper layers to preserve task-specific features. Similarly, ALD dynamically adjusts layer-dropping probabilities based on merged token counts and temporal schedules. Naively applying prior static approaches results in significant accuracy drops (Table 3), whereas these tailored mechanisms maintain accuracy while improving efficiency.
>
> In addition to these algorithmic components, REP offers a framework-level contribution by integrating them into a unified system applicable to diverse scenarios. It generalizes beyond prompting methods to non-prompting and self-supervised continual learning approaches (Sec. 4.3), consistently reducing computation and memory usage (up to 48% and 31%, respectively) with only minor accuracy degradation. REP is also instantiated on a real edge device (Appendix K), demonstrating its practical viability for on-device continual learning.
>
> We believe that by combining these algorithmic-level components with a framework-level contribution that is both generalizable and practically validated, REP represents a novel and comprehensive solution for resource-efficient continual learning.
>
> > ‘Q2: … introduce new hyperparameters …clear intuition or guidelines on how to set these parameters …without performing expensive, task-specific tuning… limit the "plug-and-play" applicability of the framework.
>
> **[Response]**
> Thank you for raising this point. Following prior work (e.g., [1]), we tuned the hyperparameters through grid search with 5-fold cross-validation for each task. While this provides an automatic tuning procedure, we agree with the reviewer that it may limit the “plug-and-play” applicability of REP. This limitation is not unique to our method and is shared by many existing continual learning approaches.
>
> Empirically, we observed that performance is relatively insensitive to variations in alpha, making it reasonable to fix alpha=0.9. For tau, the impact is more dataset- and model-dependent, but we found a consistent positive correlation between backbone size and the optimal tau value (see also our response to Reviewer mcMq, where we have a broader range of experiments). We will explicitly state this limitation in the final version and suggest a more systematic analysis of hyperparameter tuning as a future work direction.
>
>
> > ‘Q3: The paper notes that in some cases (e.g., L2P with ViT-L), REP can even improve model accuracy. The authors hypothesize that this might be because ALD and ATOM help eliminate redundant computations, effectively acting as a form of regularization. This is a very interesting phenomenon, but the paper fails to provide a deeper analysis to support this hypothesis. It is unclear whether this effect is due to the suppression of noisy features or some other mechanism. Adding some visualizations or quantitative analysis to investigate this would have significantly enhanced the paper's depth.’
>
> **[Response]**
> Thank you for noting this phenomenon. We agree that a deeper analysis would strengthen the paper. To this end, we conducted additional experiments analyzing activation variance as a proxy for noisy features in the backbone model's representations. Specifically, we measured the normalized variance of token norms in the last layer. Using Split ImageNet-R (10 tasks) with ViT-L as the backbone, we compared REP-L2P with the baseline L2P-L.
>
> | Method | Acc | Fgt | Norm. Variance |
> |------------|------------|------------|------------|
> | L2P-L | 75.6±1.0 | 2.8±0.2 | 0.29  |
> | REP-L2P | 75.3±1.5 |3.4±0.7 | 0.20 (-30%) |
>
> REP reduces average activation variance by 30%, suggesting that AToM and ALD suppress noisy features, which likely contributes to the observed accuracy improvements in certain settings. We propose to include this preliminary analysis, supporting the regularization effect introduced by REP, in the supplemental material.
>
>
>
> Reference
>
> [1] Veniat et al., Efficient continual learning with modular networks and task-driven priors, ICLR 2021.

---

> > ### Author Response · Authors · 2025-08-04
> > **Dear Reviewer**
> >
> > Dear reviewer tV8e,
> >
> > We sincerely appreciate your thoughtful review and valuable feedback, which have been invaluable in improving our research. It has been a while since we posted our last response, so we would like to follow up to see whether our response has sufficiently addressed your concerns. If there is anything else the reviewer wants us to address more, please let us know. We would be happy to engage in any further discussion.
> >
> >
> > Best regards,
> >
> > Authors

---

> ### Comment · Area_Chair_RW1S · 2025-08-05
> **post-rebuttal comments**
>
> Dear reviewer tV8e,
>
> As you may have seen already, the authors have responded to the questions in your initial review. Can you please share your thoughts post rebuttal, once you've had a chance to see the author responses and also the other reviews?
>
> Best, AC

---

> ### Comment · Reviewer_tV8e · 2025-08-09
>
> I appreciate the authors’ thorough rebuttal and find that they have satisfactorily addressed all of my concerns. I therefore uphold my original positive evaluation.

---

### Official Review · Reviewer_mcMq · 2025-07-02

**Clarity:** 4
**Significance:** 3
**Originality:** 3
**Rating:** 5
**Confidence:** 4

**Summary:**

The authors addressed the resource efficiency in continual learning with strong pretrained backbones (e.g., ViT). The authors first analyze the potential computation redundancy in prompt selection and prompt updates, and then propose solutions for each of them. For prompt selection, the authors propose to use a lightweight surrogate model and a random projection to replace the full pass on the backbone model for obtaining the query feature. For prompt update, the authors first analyze the frozen backbone to highlight the richer information contained in shallow layers. Therefore, the proposed Adaptive token Merging (AToM) preserves prompt tokens and merges more aggressively in deep layers. Similarly, the proposed Adaptive layer dropping (ALD) also drops more aggressively in deep layers. Extensive experiments showed that the proposed method can effectively reduce the computation cost with a moderate performance drop.

**Questions:**

1. The hyperparameter comparison can be improved

In the hyperparameter comparison in the Appendix (which is not referred to in the main paper, I would encourage the author to add this reference), the parameter alpha varies between 10 and 16, which is a rather small range. I am wondering how it would impact the performance if we test with a wider range.

2. The presentation of ablation can be improved

The ablation table is not easy to follow. I believe the row (1)-(6) shows the performance without the corresponding components. Please correct me if I'm understanding it wrong. First, there is no baseline performance as a reference. Second, it is unclear why (6) is more computationally efficient than (2) and (3).

3. About the prompt selection

Did the authors try to use the query feature from the intermediate layers of the f_query or f_efficient and project it with the random projection? Would this be an alternative to using a surrogate model?

Overall, I believe this is a good paper with good motivation, interesting analysis, and effective method design. It would bring insights to the community for a more resource-efficient CL.

**Ethical Concerns:**

["NO or VERY MINOR ethics concerns only"]

**Final Justification:**

I thank the authors for the detailed rebuttal and believe that they have properly addressed all my concerns. Therefore, I maintain my initial positive rating.

**Limitations:**

Yes

**Quality:**

4

**Strengths And Weaknesses:**

Strengths:

1. The paper is well motivated and addresses an interesting problem of computation efficiency

2. The paper is well-written and easy to follow

3. The proposed method is well motivated and simple, but effective

Weaknesses:

I don't see many significant weaknesses in this work. There are a few minor remarks

1. The hyperparameter comparison can be improved

2. The presentation of ablation can be improved

Please see details in the Questions section.

---

> ### Author Rebuttal · Authors · 2025-07-31
>
> We appreciate your detailed review and constructive comments, which are addressed below.
>
> > ‘Q1: The hyperparameter comparison can be improved’
>
> **[Response]**
> Thank you. We conducted experiments with a broader range of parameter values (alpha=0.6--0.95 and tau=6--20) and observed similar trends as before, as shown in the table below. The notation for alpha and tau was inadvertently swapped in Supplemental Figure 8 of our submission. We apologize for the mistake. As noted in line 682, hyperparameters were automatically tuned via five-fold cross-validation on the training set. The complete set of results will be included in the final version.
>
> | α \\ τ | 6 | 8 | 10 | 12 | 14 | 15 | 16 | 17 | 18 | 20 |
> |-------:|---:|---:|---:|---:|---:|---:|---:|---:|---:|---:|
> | 0.6  | 74.32 | 74.62 | 74.75 | 74.45 | 74.87 | 74.99 | 74.93 | 75.12 | 74.82 | 74.82 |
> | 0.7  | 74.17 | 74.38 | 74.45 | 74.98 | 74.18 | 75.06 | 75.15 | 75.26 | 74.43 | 74.31 |
> | 0.8  | 74.48 | 74.50 | 74.41 | 74.36 | 74.87 | 75.29 | 74.91 | 75.20 | 74.99 | 74.39 |
> | 0.9  | 74.49 | 74.36 | 74.77 | 74.38 | 74.58 | 75.33 | 75.34 | 75.42 | 74.88 | 74.96 |
> | 0.95 | 74.45 | 74.76 | 74.39 | 74.54 | 74.69 | 75.39 | 75.32 | 75.38 | 74.76 | 75.13 |
>
> > ‘Q2: The presentation of ablation can be improved’
>
> **[Response]**
> Thank you. The ablation results in Table 2 were reported additively from the baseline L2P-L, rather than subtractively from our final design. Our algorithm integrates three key components: (a) surrogate model f_efficient, (b) adaptive token merging (AToM), and (c) adaptive layer dropping (ALD). In Table 2, entry `(2) AToM' indicates that only AToM was used among the three components, which explains why configuration (6) is more computationally efficient. We will revise the table so that each row reflects performance with the corresponding component removed. The updated table will be as shown below.
>
> | Ablated components          | Acc. (↑) | Fgt. (↓) | GPU Time (s) | Mem. (GB) |
> |-----------------------------|---------:|---------:|-------------:|----------:|
> | Final design (REP-L2P) | 75.3±1.5 | 3.6±0.3  | 2542.4       | 4.5       |
> | (1) w/o (AToM+ALD)   | 74.8±0.8 | 4.0±0.8  | 3698.8       | 5.5       |
> | (2) w/o (f_efficient + ALD) | 74.9±0.8 | 3.5±0.8  | 4442.0       | 5.6       |
> | (3) w/o (f_efficient + AToM)  | 74.5±0.8 | 3.6±0.7  | 4244.0       | 6.7       |
> | (4) w/o  ALD | 74.2±0.8 | 4.0±0.8  | 2861.4       | 4.5       |
> | (5) w/o  AToM | 74.5±0.4 | 3.4±0.5  | 3206.7       | 5.5       |
> | (6) w/o f_efficient | 74.6±0.4 | 2.6±0.5  | 3448.8       | 4.8       |
>
>
> > ‘Q3: About the Prompt Selection’
>
> **[Response]**
> Thank you for the suggestion. We had not initially tested this approach, but your question prompted us to conduct experiments, with results shown in the table below. Specifically, we evaluated prompt selection using CLS tokens from ViT-L’s intermediate layers (l = 4, 8, 12). While using intermediate layers reduces the need for a separate surrogate, it increases GPU time since even a partial ViT-L forward pass (to l=8, covering ~67% of layers) is more computationally expensive than running the full ViT-Ti (5M vs. 307M params). Accuracy also drops for shallower layers (e.g., l=4: -1.3%) due to reduced global context. These findings demonstrate the efficiency advantage of our surrogate model. We will include this analysis in the final version.
>
> | Method | Acc | Fgt | GPU Time (s) | Mem (GB) |
> |------------|------------|------------|------------|------------|
> | Final design (surrogate model) | 75.3±1.5 | 3.6±0.3 | 2542.4 | 4.5 |
> | ViT-L interm. layers (l=4) | 74.0±0.8 | 3.9±0.9 | 2792.9 | 4.8 |
> | ViT-L interm. layers (l=8) | 74.3±0.7 | 3.5±0.9 | 3022.8 | 4.8 |
> | ViT-L interm. layers (l=12) | 74.6±0.7 |2.6±0.8 | 3253.4 | 4.8 |

---

> > ### Author Response · Authors · 2025-08-04
> > **Dear Reviewer**
> >
> > Dear reviewer mcMq,
> >
> > We sincerely appreciate your thoughtful review and valuable feedback, which have been invaluable in improving our research. It has been a while since we posted our last response, so we would like to follow up to see whether our response has sufficiently addressed your concerns. If there is anything else the reviewer wants us to address more, please let us know. We would be happy to engage in any further discussion.
> >
> >
> > Best regards,
> >
> > Authors

---

> > ### Comment · Reviewer_mcMq · 2025-08-06
> >
> > I thank the authors for the detailed rebuttal and believe that they have properly addressed all my concerns. Therefore, I maintain my initial positive rating.

---

> ### Comment · Area_Chair_RW1S · 2025-08-05
> **post-rebuttal comments**
>
> Dear reviewer mcMq,
>
> As you may have seen already, the authors have responded to the questions in your initial review. Can you please share your thoughts post rebuttal, once you've had a chance to see the author responses and also the other reviews?
>
> Best, AC

---

### Note · Authors · 2025-08-13

We thank our reviewers for their constructive feedback, which helped us examine our work from diverse perspectives and provided concrete suggestions for improving the paper. We will incorporate all review comments, including the following specific aspects:

- Reviewer mcMq's suggestions on refining our hyperparameter choices and clarifying our ablation studies.

- Reviewer tV8e's comments on deepening our analysis of REP's novelty and its potential regularization effects.

- Reviewer jSzb's feedback on the efficiency–accuracy trade-off and the potential of integrating Fisher Information Matrix-based optimization.

- Reviewer W2oY's observations leading us to expand our trade-off mapping and clarify curriculum-related considerations.

We conducted additional experiments on broader hyperparameter grids, prompt selection under diverse domain shifts, and static acceleration baselines versus REP, confirming REP’s value as a resource-efficient approach for on-device continual learning scenarios.

We took particular note of Reviewer jSzb’s concern regarding the trade-off between performance drop and computational gain. Our work, like many in efficient continual learning, does not aim to increase accuracy but to provide a tunable trade-off between performance and resource efficiency (e.g., memory). In many practical CL scenarios, including edge environments, modest accuracy trade-offs are both acceptable and necessary.

The reviewer also asked whether the trade-off is worthwhile across all settings. While a system that excels in every case would be ideal, we respectfully suggest that this expectation may set an unrealistic bar. No method performs optimally across all settings when navigating inherent trade-offs. REP consistently delivers substantial computational savings, with typical accuracy drops under 1%, and we have transparently reported rare cases of larger drops (~2%). We kindly ask that the reviewers consider these results in the context of real-world constraints, where efficiency gains can outweigh minor accuracy reductions.

Sincerely yours,

The authors

---

### Decision · Program_Chairs · 2025-09-17

**Decision:**

Accept (poster)

**Comment:**

This paper aims to improve the computational and memory efficiency of rehearsal-free continual learning. In particular, it targets the prompt-based rehearsal-free continual learning paradigm. The proposed approach is based on insights made in the prompt selection and the prompt update stages. This paper then proposes techniques to improve the prompt selection process and also adaptive layer dropping and token merging to improve the prompt update process.

During the review process, the strengths of the paper were identified as: exceptionally thorough evaluation, a framework that delivers substantial resource-efficiency, being adaptable and modular (owing to the diversity of the tasks, models, datasets considered), etc.

After the reviewer-author and the reviewer-AC discussion phases, during which all the reviewers participated actively, the reviewers acknowledged that the questions in their initial reviews were well-addressed. The only major point was reviewer jSzb's comment on whether the trade-off between performance drop and resource efficiency is worthwhile. The AC notes that this point was well treated in the author responses, which clearly show both the positive and the negatives sides to this trade-off. Given this balanced presentation and the overwhelming positive evaluations of all the reviewers (including reviewer jSzb), the AC recommends the paper for acceptance.

The final version should include all the clarifications provided in the rebuttal and during the discussion phase.